# Spatiotemporal Analysis of Emergency Calls during the COVID-19 Pandemic: Case of the City of Vaughan

Ali Asgary [1,]*, Adriano O. Solis [2], Nawar Khan [1], Janithra Wimaladasa [1] and Maryam Shafiei Sabet [3]

1. Disaster & Emergency Management Area, School of Administrative Studies, York University, Toronto, ON M3J 1P3, Canada; nawarkhan13@gmail.com (N.K.); janithrac@yahoo.com (J.W.)
2. Decision Sciences Area, School of Administrative Studies, York University, Toronto, ON M3J 1P3, Canada; asolis@yorku.ca
3. Fleming College, Sutherland Campus, Peterborough, ON K9K 2N7, Canada; maryamshsabet@gmail.com
* Correspondence: asgary@yorku.ca

**Abstract:** Cities have experienced different realities during the COVID-19 pandemic due to its impacts and public health measures undertaken to respond to and manage the pandemic. These measures revealed significant implications for municipal functions, particularly emergency services. The aim of this study is to examine the spatiotemporal distribution of emergency calls during different stages/periods of the pandemic in the City of Vaughan, Canada, using spatial density and the emerging hotspot analysis. The Vaughan Fire and Rescue Service (VFRS) provided the dataset of all emergency calls responded to within the City of Vaughan for the period of 1 January 2017 to 15 July 2021. The dataset was divided according to 11 periods during the pandemic, each period associated with certain levels of public health restrictions. A spatial analysis was carried out by converting the data into shapefiles using geographic coordinates of each call. Study findings show significant spatiotemporal changes in patterns of emergency calls during the pandemic, particularly during more stringent public health measures such as lockdowns and closures of nonessential businesses. The results could provide useful information for both resource management in emergency services as well as understanding the underlying causes of such patterns.

**Keywords:** spatiotemporal analysis; COVID-19; pandemic; emergency calls; kernel density analysis; emerging hotspot analysis; city of Vaughan; Vaughan fire and rescue service (VFRS)





## 1. Introduction and Background

Cities have experienced different realities during the COVID-19 pandemic. The pandemic and public health measures significantly altered urban functions and operations. Since the beginning of the COVID-19 pandemic in Canada in March 2020, the City of Vaughan has gone through several waves of the pandemic, each associated with specific stages of public health measures. Physical distancing measures, in particular, have required full or partial closure of public facilities, schools, non-essential businesses, and places of worship, among others. These measures could have significant implications for municipal functions, particularly emergency services. Emergency calls that are mainly responded to by the city's fire and rescue services started to change in terms of composition and frequency during the pandemic. The aim of this study is to examine the spatiotemporal distribution of emergency calls during different phases/periods of the pandemic in the City of Vaughan using spatial density and the emerging hotspot analysis.

In cities and municipalities in the province of Ontario, Canada, emergency calls that are responded to by fire departments, referred to in certain cases as fire and rescue services, are not restricted to fires or fire-related emergencies. Emergency incidents fall under a number of major categories other than property fires/explosions, including false fire calls, medical emergencies, vehicle collisions/extractions, public hazards (including carbon monoxide), and others. Emergency incidents falling under various categories will call for different

resource requirements, e.g., the number and types of responding vehicles and the number and training/experience of crew members. The geographical expanse of a city/municipality and the distribution of properties across the property type (e.g., residential, business and personal services, industrial, mercantile, assembly, care and detention, vehicles, etc.) would usually call for the subdivision of the city/municipality into fire districts and the distribution or allocation of firefighting and the rescue vehicles/apparatus across different fire stations.

A spatiotemporal analysis can play a vital role in understanding and analysing data with spatial attributes. Spatiotemporal modelling methodologies are rapidly developing and evolving. The emerging hotspot analysis is among the new methods added to the GIS-based analyses and its usage in a spatiotemporal analysis is growing [1–7]. Gudes et al. [1] used spatial modelling and spatiotemporal methods to identify emerging hotspots of heavy-vehicle crashes on specific roads in Western Australia. Rabiei-Dastjerdi and McArdle [2] investigated patterns of neighbourhood change by using EHA of Airbnb data in the City of Dublin, Ireland. Using EHA, Reddy et al. [3] found the dominance of sporadic hotspots and persistent hotspots in vegetation fire occurrences in Myanmar and South Asian countries. EHA appears to have been used prevalently in forecasting crimes, e.g., by Hart [4] to forecast crime hotspots for three types of crime handled by six USA law enforcement agencies, by Adepeju et al. [5] for three crime types in the borough of Camden (London, UK) and South Chicago (Chicago, IL, USA), by Mohler [6] for homicide and gun crimes in Chicago (IL, USA), and by Chainey et al. [7] for four crime types.

Spatial and spatiotemporal analysis tools have become indispensable in studying public health concerns during the COVID-19 pandemic. A number of studies have attempted to explore the spatial and spatiotemporal patterns, including EHA, of COVID-19 cases and underlying risk factors of COVID-19 in different contexts [8–11]. Mollalo et al. [8] applied spatial modelling tools, including a multiscale geographically weighted regression (MGWR) analysis, to the county-level counts of COVID-19 cases from 22 January to 9 April 2020 across the continental USA. They found that MGWR could explain 68% of the total variations of COVID-19 incidence. Mylona et al. [9] extracted and performed a hotspot analysis of influenza cases (2016–2019) as well as COVID-19 cases (March–April 2020) from a Rhode Island (USA) hospital network to simulate a real-time surveillance scenario. Andersen et al. [10] analysed spatial determinants of local COVID-19 transmission in the USA and found COVID-19 hotspots predominantly in New England, southeast, and southwest states. Purwanto et al. [11] conducted a spatiotemporal analysis of the COVID-19 spread in East Java, Indonesia, using EHA and space–time cube models. Results showed that the spread of COVID-19 in East Java was centred in Surabaya, then spread to suburban areas and other cities.

Considering that the City of Vaughan had been under a widespread COVID-19 emergency, understanding how this emergency impacts the overall spatial and spatiotemporal distribution of specific types of emergencies during different phases and waves of the pandemic can help decision makers better adapt their resources and be aware of the situation as public health measures change over time [12]. This information would be particularly useful when the calls related to COVID-19 and other emergency calls overlap and understanding such patterns to minimize the impacts on most vulnerable groups [13]. Use of spatiotemporal analysis methods in emergency calls and incidents is not new. However, as technology advances, new methods have been developed and applied, including the Exploratory Spatial Data Analysis, Parallel Coordinates Plot (PCP), Multiform Bivariate Matrix, SpaceFill Visualisation, Geographically Weighted Regression, and Spatial Machine Learning [13–18].

The present study applies a number of spatial and spatiotemporal methods to examine patterns of emergency calls in Vaughan during the first 11 phases of the COVID-19 pandemic (spanning the first 16 months, as summarized in Section 2.2) in the City of Vaughan, Ontario, Canada. In particular, we were interested in finding answers to the following questions: how has the pandemic changed the emergency calls in the city? Has this pattern

changed during different phases of the pandemic under different public health measures? Which emergency calls had the highest and lowest spatial–temporal changes? In an earlier study, Solis et al. [19] had applied temporal data analytics methods to investigate changes in number and nature of emergency incidents in Vaughan through the first six stages, covering roughly the first ten months, of the pandemic and associated public health measures.

This study contributes to the literature by being among the first studies to examine such spatiotemporal aspects for more than one year of the pandemic under different levels of public health measures. It offers emergency service providers such as the VFRS additional information about the spatial and spatiotemporal trends of emergency calls, which is crucial to planning and management of resources under the ongoing pandemic.

## 2. Datasets and Methods

The workflow of this study consists of four steps (Figure 1): data collection and preparation for emergency calls from the City of Vaughan Fire and Rescue Service (1), creating the geodatabase combining base layers and the emergency call layers for different years and pandemic phases (2), the spatiotemporal pattern analysis of emergency calls (3), and resource allocation planning and applications (4).

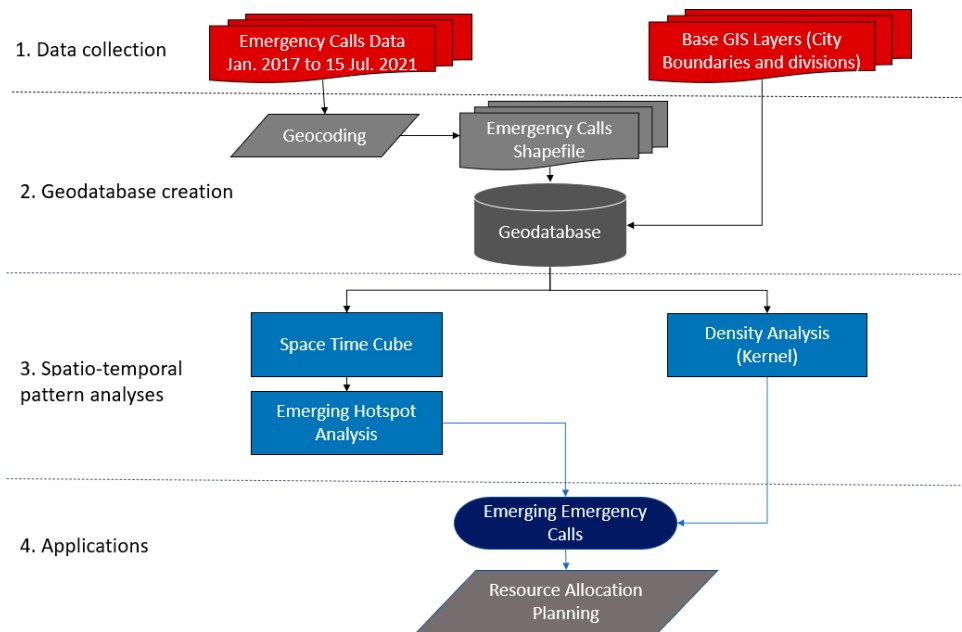

**Figure 1.** Study workflow.

### 2.1. Study Area

This study uses data collected from the City of Vaughan, which is one of nine municipalities in the Regional Municipality of York (or York Region) of the Canadian province of Ontario. Vaughan is situated north of the City of Toronto (Figure 2a), the capital of Ontario. Its population in 2021 was estimated to be around 335,000 and is expected to grow to 575,500 people by 2031 [18].

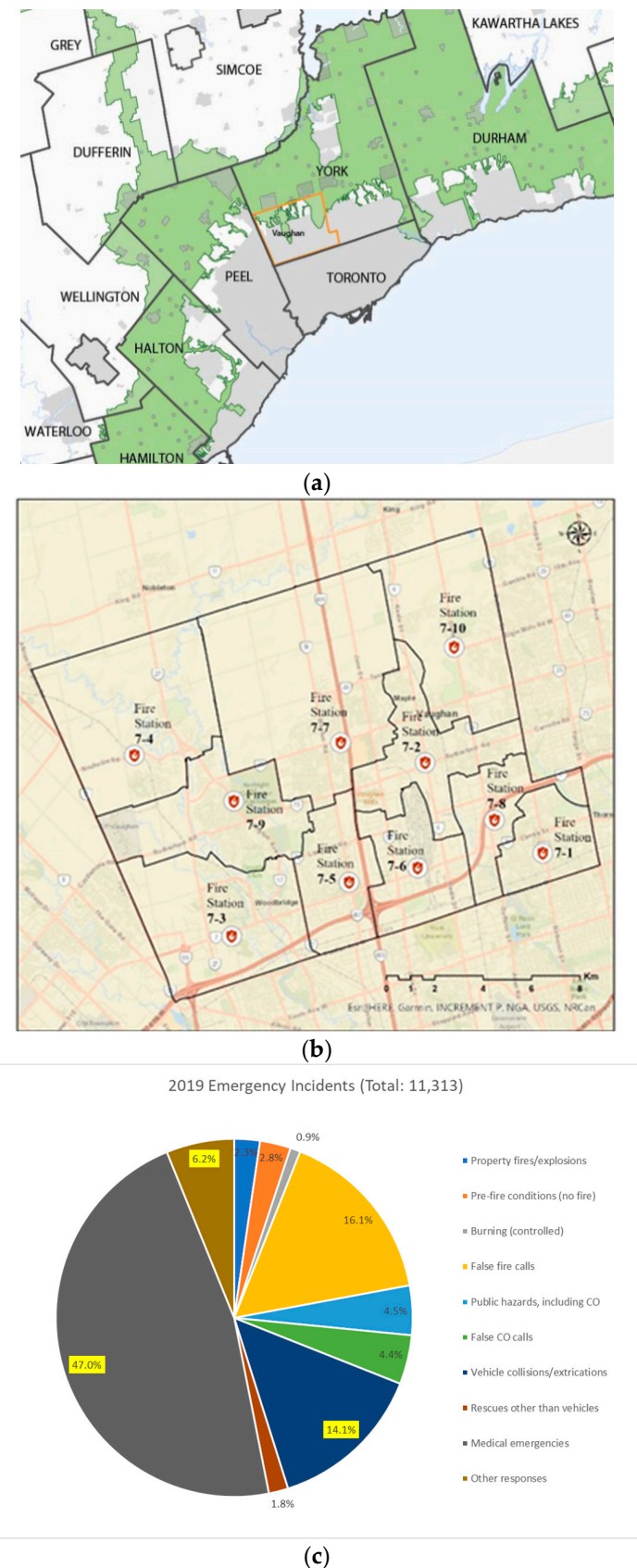

**Figure 2.** (**a**) Location of the City of Vaughan [18]. (**b**) Fire districts and fire stations in the City of Vaughan as of January 2020. (**c**) Percentage distribution of emergency incidents, by category, in the City of Vaughan in 2019.

The Vaughan Fire and Rescue Service (VFRS) is the city's emergency response organization. It currently operates with 10 fire districts (71, 72, . . . , 79, and 710) (Figure 2b) with corresponding stations (7-1, 7-2, . . . , 7-9, and 7-10), with the former District 79 split into two, and with the addition of District 74 in January 2020. VFRS provides full-service emergency response for fire incidents, medical emergencies, vehicular and non-vehicular rescues, hazardous material incidents (e.g., chemical, biological, radiological, and nuclear), and others [18].

As of January 2020, two responding units—where the term 'responding unit' refers to the firefighting apparatus manned by a crew of four firefighters—were stationed at each of the VFRS' Stations 7-1, 7-2, and 7-3. The seven other fire stations each have only one responding unit. However, in addition to one full responding unit, Station 7-5 also has a first response unit, which has a crew of at least two firefighters using a pick-up truck, mostly responding to medical emergency calls. A limited number of the specialized firefighting apparatus is based at designated fire stations. For example, engines having 100 ft. aerial equipment are based at Stations 7-1 and 7-3, with more high-rise buildings located in Districts 71 and 73 than in other districts.

In 2019, the calendar year immediately preceding the declaration of the COVID-19 pandemic, the VFRS responded to a total of 11,313 emergency incidents of various types (see Figure 2c). Medical emergencies accounted for 47% of all emergency incidents, followed by false fire calls (16.1%) and vehicle collisions/extrications (14.1%). There were only 263 property fires/explosions, accounting for only 2.3% of all emergency incidents in 2019.

### 2.2. Data Collection and Preparation

VFRS provided the dataset of all occurrences of incidents within the City of Vaughan for the period of 1 January 2017 to 15 July 2021 through a non-disclosure agreement. Incident data during the pandemic were provided incrementally after the end of each period of the pandemic. Each incident involved attributes including incident number, latitude, longitude, alarm date and time, station, district, zone, incident (response) type, dispatch date and time, arrival date and time, clearing date and time, and alarm type and property type, as specified in a Standard Incident Report (SIR) Codes List issued by the Office of the Fire Marshal of Ontario. For the analysis, the dataset was divided according to the reference periods during the COVID-19 pandemic. The spatial analysis was carried out converting the data into shapefiles using the records of latitude and longitude coordinates. Incidents with no records of spatial attributes (latitudes and longitudes) were ignored for the spatial and spatiotemporal analyses.

The first 11 periods of the COVID-19 pandemic for the City of Vaughan are summarized in Table 1. These 11 periods are based upon public health measures introduced in the province of Ontario and in York Region. As a result of the declaration of the first State of Emergency and its associated orders, certain establishments were legally required to close immediately, while all organized public events of over 50 people were also prohibited, including parades, events, and communal services within places of worship [20].

**Table 1.** Pandemic periods/phases for the City of Vaughan, Ontario.

| Period | No. of Days | Brief Description of Period | Reference(s) |
| --- | --- | --- | --- |
| Period 1 (17 March–18 May 2020) | 63 | State of Emergency I (lockdown began) | [20] |
| Period 2 (19 May–18 June 2020) | 31 | Stage 1 reopening | [21] |
| Period 3 (19 June–23 July 2020) | 35 | Stage 2 reopening | [21] |
| Period 4 (24 July–18 October 2020) | 87 | Stage 3 reopening | [22] |

**Table 1.** *Cont.*

| Period | No. of Days | Brief Description of Period | Reference(s) |
|---|---|---|---|
| Period 5 (19 October–13 December 2020) | 56 | Modified Stage 2 reopening | [23] |
| Period 6 (14 December 2020–13 January 2021) | 31 | York Region lockdown | [24,25] |
| Period 7 (14 January–21 February 2021) | 39 | State of Emergency II: Stay-at-home order | [26] |
| Period 8 (22 February–2 April 2021) | 40 | York Region as a 'red zone' | [27] |
| Period 9 (3 April–10 June 2021) | 69 | 'Stay-at-home' order; initially 'emergency brake' order | [28,29] |
| Period 10 (11 June–29 June 2021) | 19 | Step 1 of Province of Ontario's Roadmap to Reopen | [30] |
| Period 11 (30 June–15 July 2021) | 16 | Step 2 of Province of Ontario's Roadmap to Reopen | [31] |

The reference dates as specified for Periods 2–5 are associated with stages of reopening of the economy applying to York Region [21–23]. Period 6 pertains to York Region being placed in the Grey/lockdown zone of the Keeping Ontario Safe and Open COVID-19 Response Framework [24,25]. Period 7 involves a second State of Emergency for Ontario [26]. Period 8 pertains to York Region being declared a Red/control zone [27]. Period 9 initially involved an 'emergency brake' order taking effect on 3 April 2021 [28], which was very quickly superseded by a 'stay-at-home' order starting on 8 April 2021 [29]. Ontario's Roadmap to Reopen (Ontario Regulation 363/20) identifies which restrictions are lifted under Steps 1, 2, and 3 [30,31].

*2.3. Methods*

To analyse the data on emergency incidents, we first performed a kernel density analysis using ArcGIS Pro 2.8 to examine changes in the spatial distribution of emergency calls (total and by major incident type) before and during various periods/stages of the pandemic. The kernel density tool calculates the density of point features around each unit of space (i.e., 10 m × 10 m) output raster cell based on a quartic kernel function [32].

Subsequently, we applied the emerging hotspot analysis (EHA) using ArcGIS Pro software [33] to understand spatiotemporal variations of emergency calls during the study period. EHS uses Mann–Kendall statistics to detect and examine statistically significant trends. EHA is able to provide a summary of spatial distribution, identify significant clusters in the dataset, and explore patterns over time through regression.

EHA classifies the data into several patterns including: (1) 'no pattern' when the result does not exhibit any hot- or cold-spot patterns; (2) 'new pattern' when the most recent data exhibits a statistically significant hotspot that had not previously been a significant hotspot; (3) 'oscillating pattern' when data exhibits a statistically significant hotspot in areas that have previously exhibited a statistically significant cold spot; and (4) 'sporadic pattern' when a location varies as a hotspot [1] (see Table 2). EHA calculates the *z*-score, *p*-value, and hotspot classification (none, new, oscillating, or sporadic) for each location and uses 3D visualisation to present the patterns.

**Table 2.** Definitions of Patterns (Emerging Hotspot Analysis).

| Pattern | Pattern Name | Pattern | Pattern Name |
|---|---|---|---|
| | New Hotspot | | New Cold Spot |
| | Consecutive Hotspot | | Consecutive Cold Spot |
| | Intensifying Hotspot | | Intensifying Cold Spot |
| | Persistent Hotspot | | Persistent Cold Spot |
| | Diminishing Hotspot | | Diminishing Cold Spot |
| | Sporadic Hotspot | | Sporadic Cold Spot |
| | Oscillating Hotspot | | Oscillating Cold Spot |
| | Historical Hotspot | | Historical Cold Spot |
| | No Pattern Detected | | |

Source: ESRI [33].

Analyses were carried out using emergency incident data spanning from 17 March 2019 (a year before the pandemic) to 15 July 2021 (end of Period 11). Different ranges of time periods were considered to understand the spatiotemporal changes in emergency calls before, after, and during different stages of restrictions of the pandemic. The emergency incidents were aggregated into defined locations and defined time intervals to carry out the analysis. For this purpose, space–time cubes were created by aggregating points in ArcGIS Pro. The incidents were aggregated into a hexagonal grid with a distance interval of 1 km and a time interval of 2 weeks. The 'dispatch date' recorded for the incidents in the VFRS database was considered to aggregate data into time intervals. Created space–time cubes were then used to carry out the emerging hotspot analysis. The neighbourhood distance considered for the analysis was 2 km.

## 3. Analyses and Findings

### 3.1. Density Analyses/Findings

3.1.1. All Emergency Calls

Table 3 shows the total numbers of emergency calls received by VFRS in each of the City of Vaughan's Periods 1–11 of the COVID-19 pandemic (as earlier summarized in Table 1). Except for Period 9, the total number of emergency calls during each COVID-19 period was less than the average for the same period in 2017–2019.

**Table 3.** All emergency incidents during the City of Vaughan's COVID-19 Periods 1–11 versus corresponding periods in 2017–2019.

| Period | No. of Days | 2017 | 2018 | 2019 | Average 2017–2019 | During COVID-19 |
|---|---|---|---|---|---|---|
| Period 1 (17 March–18 May 2020) | 63 | 1933 | 2013 | 1821 | 1922.3 | 1476 |
| Period 2 (19 May–18 June 2020) | 31 | 989 | 1073 | 974 | 1012.0 | 873 |
| Period 3 (19 June–23 July 2020) | 35 | 1094 | 1157 | 1139 | 1130.0 | 986 |
| Period 4 (24 July–18 October 2020) | 87 | 2790 | 2829 | 2705 | 2774.7 | 2466 |
| Period 5 (19 October–13 December 2020) | 56 | 1699 | 1822 | 1694 | 1738.3 | 1570 |
| Period 6 (14 December 2020–13 January 2021) * | 31 | 1223 | 861 | 957 | 1013.7 | 854 |
| Period 7 (14 January–21 February 2021) | 39 | 1084 | 1208 | 1363 | 1218.3 | 1044 |
| Period 8 (22 February–2 April 2021) | 40 | 1196 | 1153 | 1171 | 1173.3 | 1104 |
| Period 9 (3 April–10 June 2021) | 69 | 2122 | 2292 | 2099 | 2171.0 | 2236 |
| Period 10 (11 June–29 June 2021) | 19 | 654 | 628 | 589 | 623.7 | 549 |
| Period 11 (30 June–15 July 2021) | 16 | 501 | 517 | 515 | 511.0 | 484 |

Note: * For Period 6, numbers reported for 2017, 2018, and 2019 are for 14 December 2017–13 January 2018, 14 December 2018–13 January 2019, and 14 December 2019–13 January 2020, respectively.

Figure 3 illustrates the average number per day of all emergency calls received by VFRS in each of Periods 1–11 of COVID-19 in comparison with corresponding average numbers in the three pre-pandemic years (2017, 2018, and 2019). Except in Period 9, the average number in each period of the pandemic was lower than the average numbers in 2017–2019. However, the average number of emergency incidents rose to 32.4 per day during Period 9 of the pandemic from 31.5 per day on average in 2017–2019. This was largely brought about by an increase in medical emergencies from 14.4 per day on average in 2017–2019 to 18.4 per day during Period 9, when there was a 'stay-at-home' order.

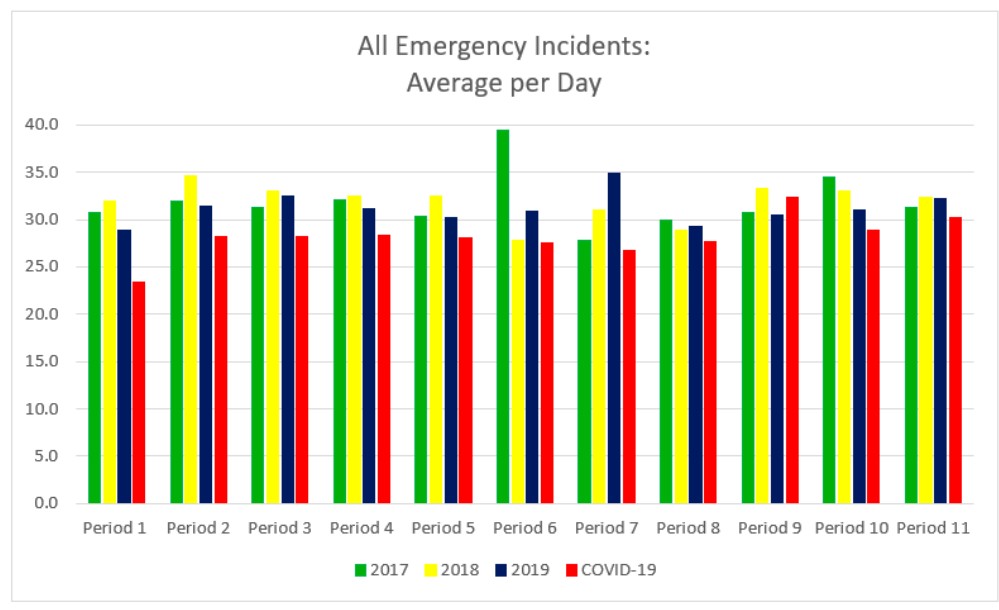

**Figure 3.** All emergency incidents: Average per day during each COVID-19 period versus corresponding periods in 2017, 2018, and 2019. Note: For Period 6, numbers reported for 2017, 2018, and 2019 are for 14 December 2017–13 January 2018, 14 December 2018–13 January 2019, and 14 December 2019–13 January 2020, respectively.

Figure 4 presents the kernel density analysis results for all types of emergency calls based on different COVID-19 pandemic periods, in comparison with corresponding periods in the 3 years (2017–2019) preceding the pandemic. Comparing density maps for the pandemic periods clearly shows significant differences between different pandemic periods. Periods with more public health measures are very different from periods in which public health measures are relaxed.

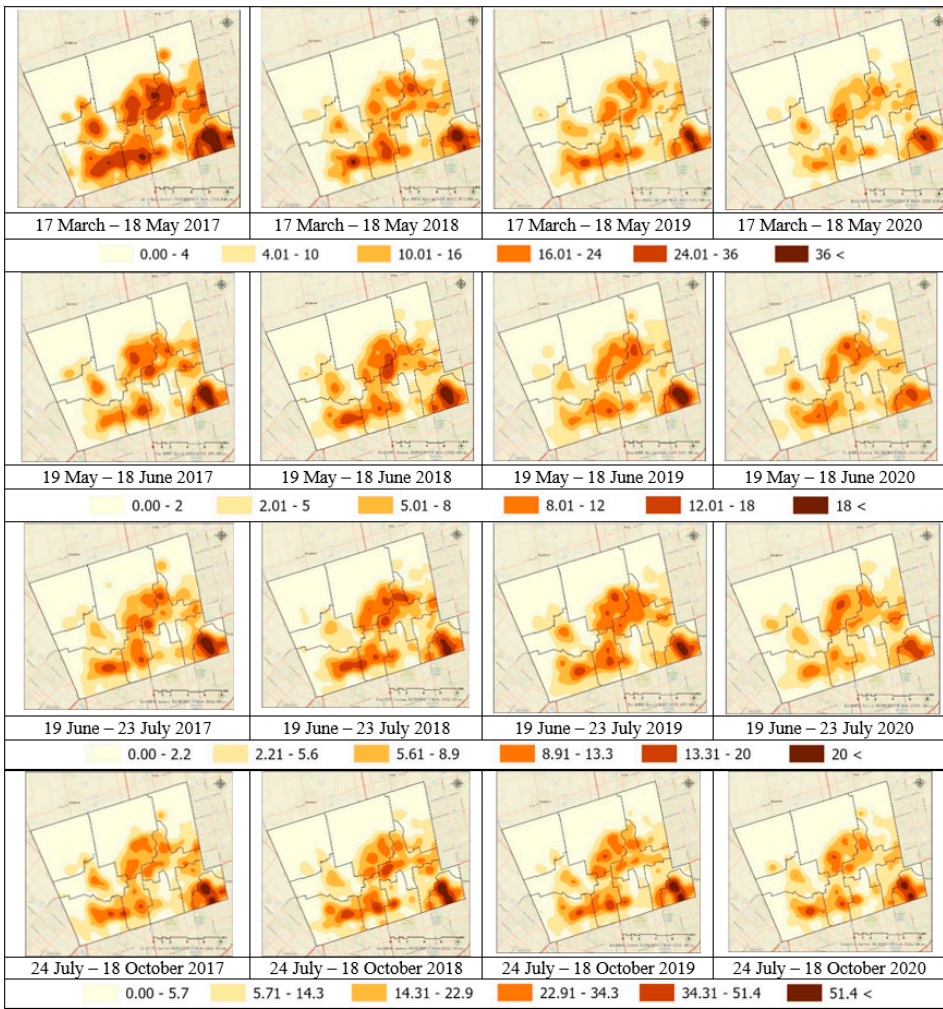

**Figure 4.** *Cont.*

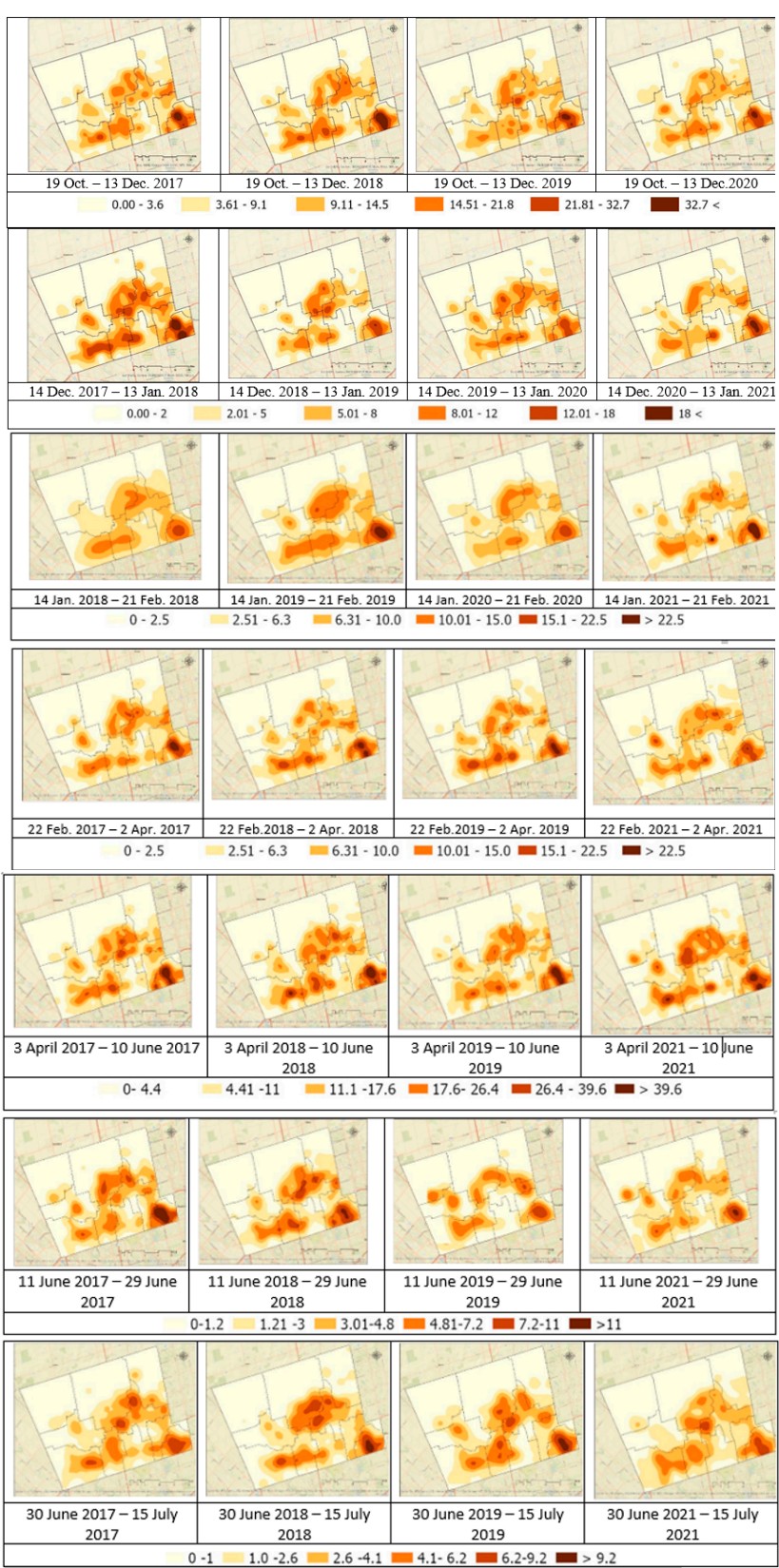

**Figure 4.** Kernel density maps of all emergency incidents during Period 4, 24 July–18 October 2017–2020 (87 days).

### 3.1.2. Vehicle Collisions/Extrications

Figure 5 illustrates the average number of medical emergencies per day in each of Periods 1–11 in comparison with average daily numbers during the same period in the 3 years preceding the pandemic (2017–2019). Average numbers of medical emergencies per day were lower during most COVID-19 periods compared to corresponding periods in 2017–2019, except in Periods 6, 8, 9, and 10, when there were more incidents per day, on average, during the pandemic period. This is most particularly evident in Period 9, when the government of Ontario issued a 'stay-at-home' order.

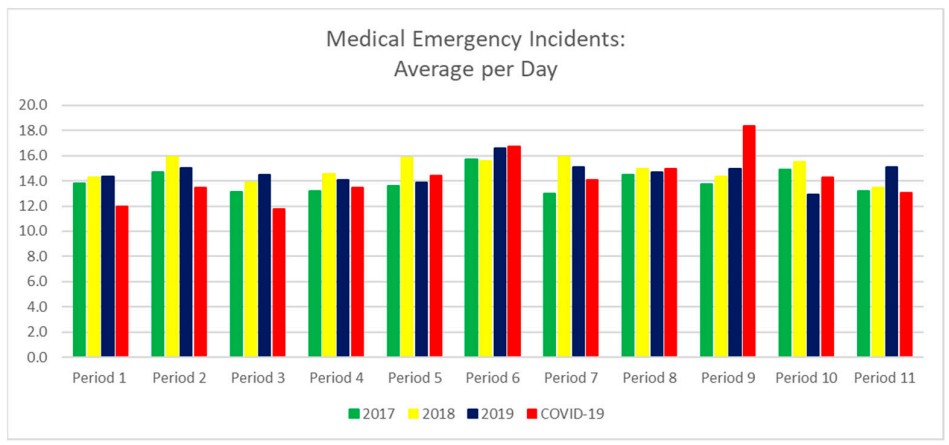

**Figure 5.** Medical emergency incidents: Average per day during each COVID-19 period versus corresponding periods in 2017, 2018, and 2019. Note: For Period 6, numbers reported for 2017, 2018, and 2019 are for 14 December 2017–13 January 2018, 14 December 2018–13 January 2019, and 14 December 2019–13 January 2020, respectively.

In this section, we present the results of the kernel density analysis for medical emergency calls during different periods of the pandemic (Figure 6). These maps show significant variations in time and space. For example, these results show significant decreases in density within Districts 71, 72, 73, and 75 (refer back to Figure 2 for the geographic areas of the 10 VFRS districts) compared to the previous year. Most significant decreases in density can be observed in District 71. Again, the results show that public health restrictions that closed non-essential businesses and public spaces influenced the density of medical emergencies.

### 3.1.3. Medical Emergencies

The average number of vehicle collisions/extrications per day during each of Periods 1–11 of the pandemic is shown alongside corresponding daily averages prior to the pandemic in 2017–2019 in Figure 7. Clearly, the occurrences of vehicle collisions/extrications have dramatically dropped in every period of the pandemic, consistent with lockdowns, 'stay-at-home' orders, and other public health measures that led to students attending classes remotely or people working from home.

Figure 8 presents the kernel density maps for vehicle collisions/extrications for 2019 and 2020 for each pandemic period. These maps clearly show less colour/lighter shades in the map for a given pandemic period relative to the map for the same period in the preceding year. This observation corresponds to a reduction in trips imposed by public health measures. One can easily observe that the vehicle collision/extrication call densities are highly correlated with the increase/decrease in public health measures during the pandemic period. As vehicle collisions occur on road networks and particularly highways that pass through certain districts, the change in these events significantly impacts fire stations in districts such as Districts 75, 76, and 73 (along Ontario Highways 7 and 407) and in Districts 77 and 72 (along Ontario Highway 400).

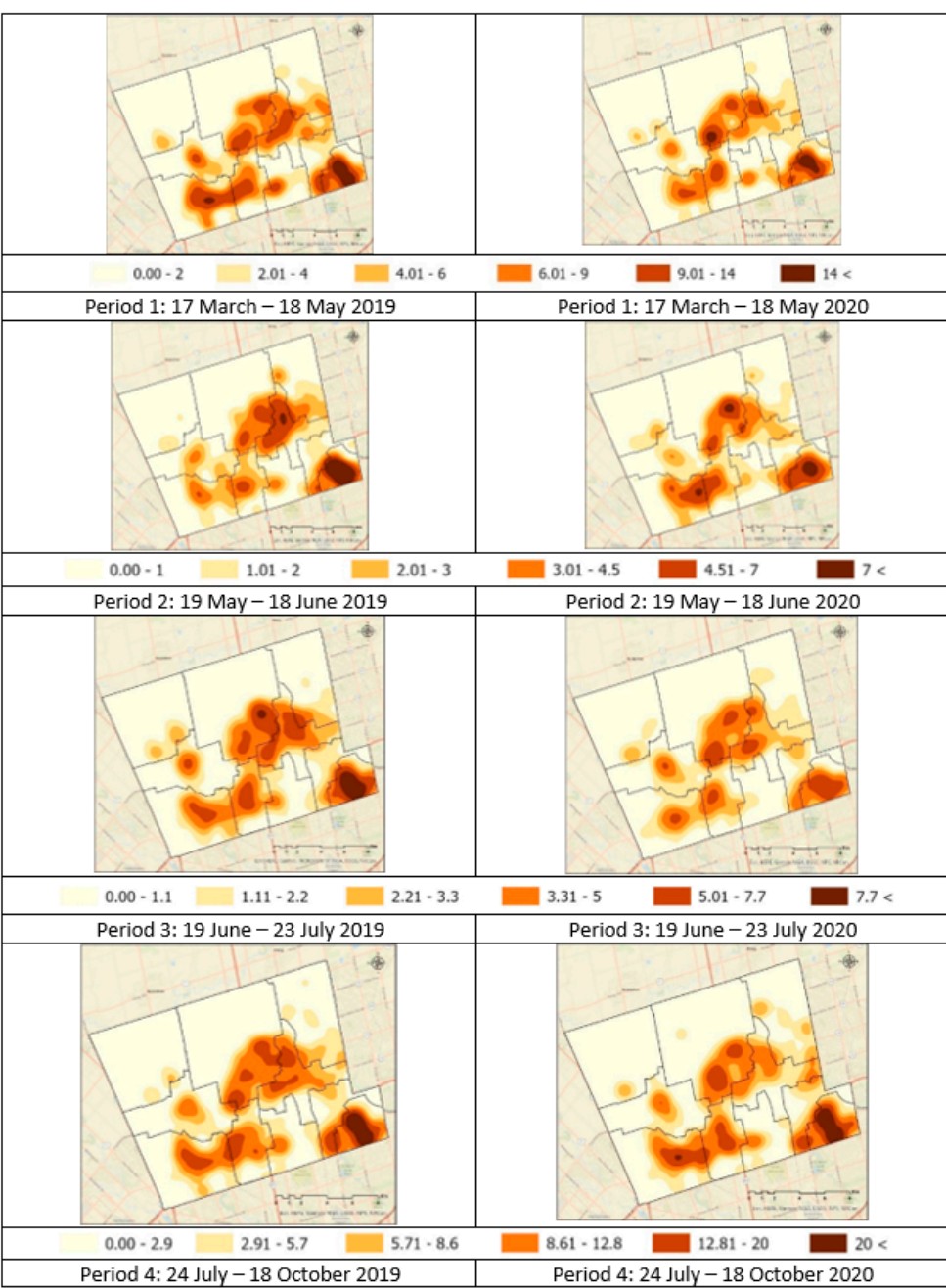

**Figure 6.** *Cont.*

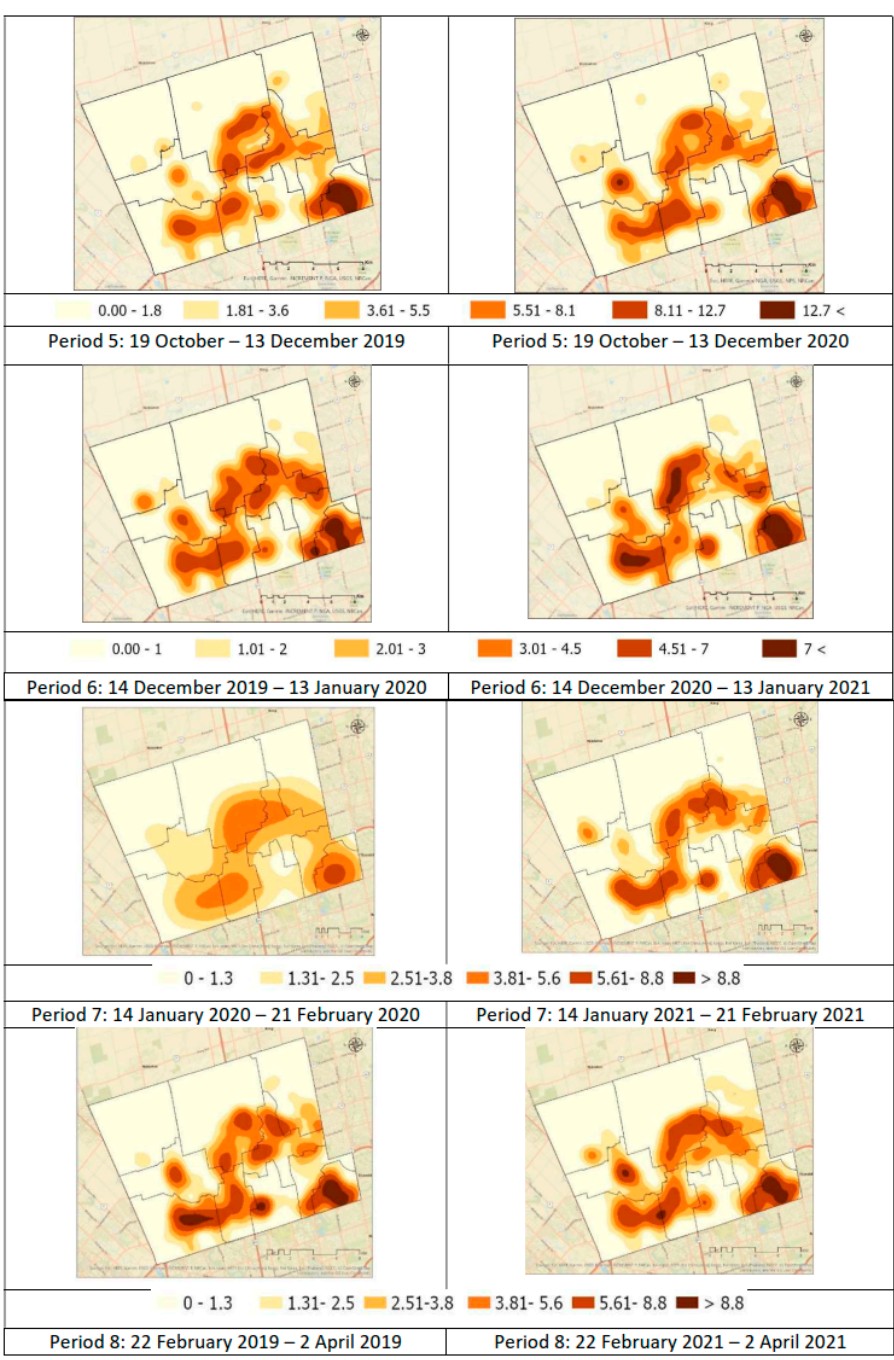

**Figure 6.** *Cont.*

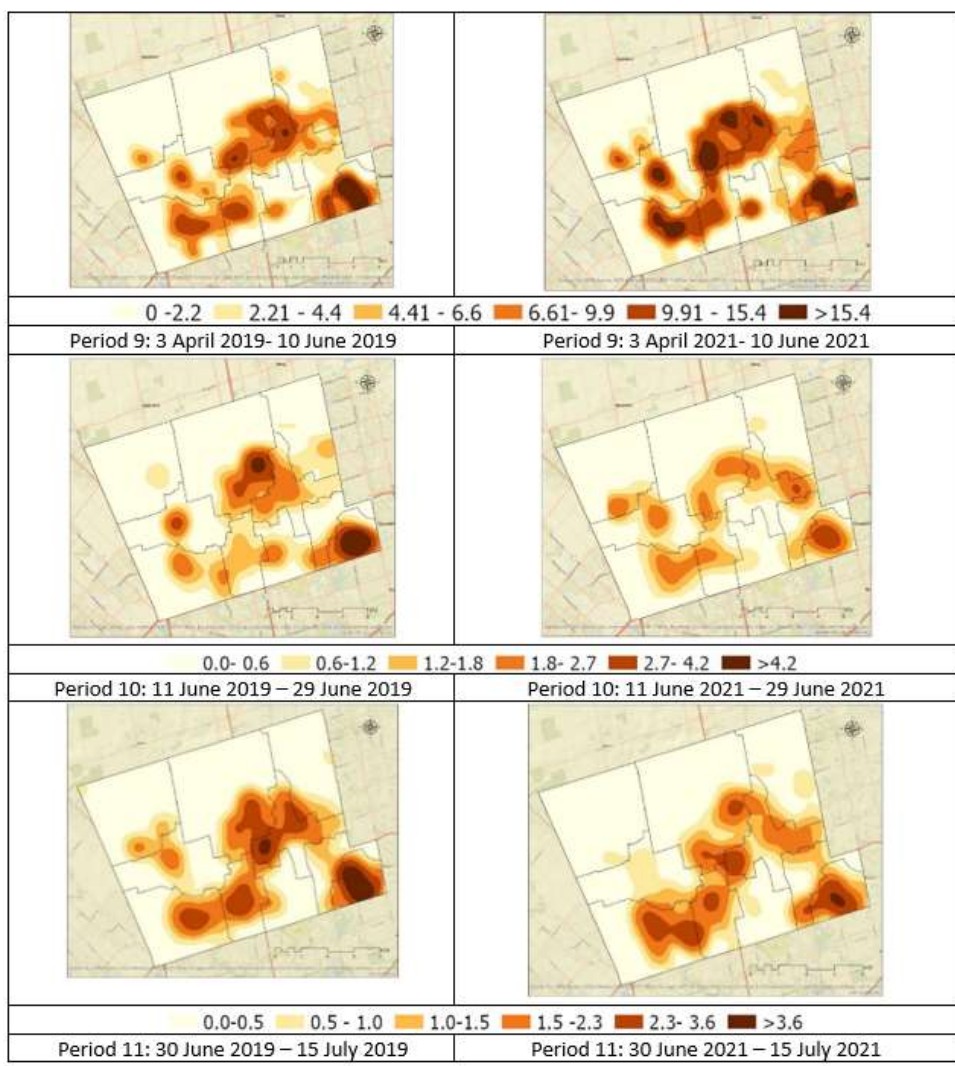

**Figure 6.** Kernel density maps for medical emergencies for Periods 1–4.

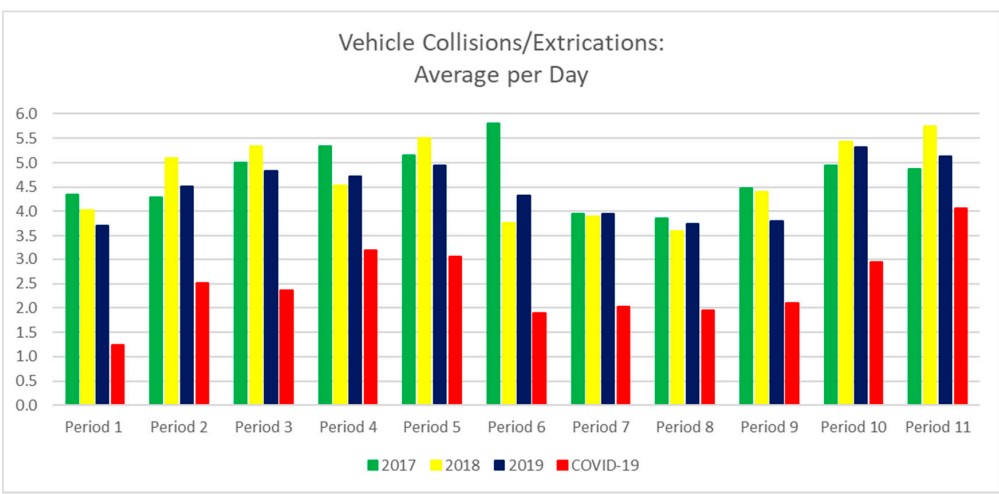

**Figure 7.** Vehicle collisions/extrications: Average per day during each COVID-19 period versus corresponding periods in 2017, 2018, and 2019. Note: For Period 6, numbers reported for 2017, 2018, and 2019 are for 14 December 2017–13 January 2018, 14 December 2018–13 January 2019, and 14 December 2019–13 January 2020, respectively.

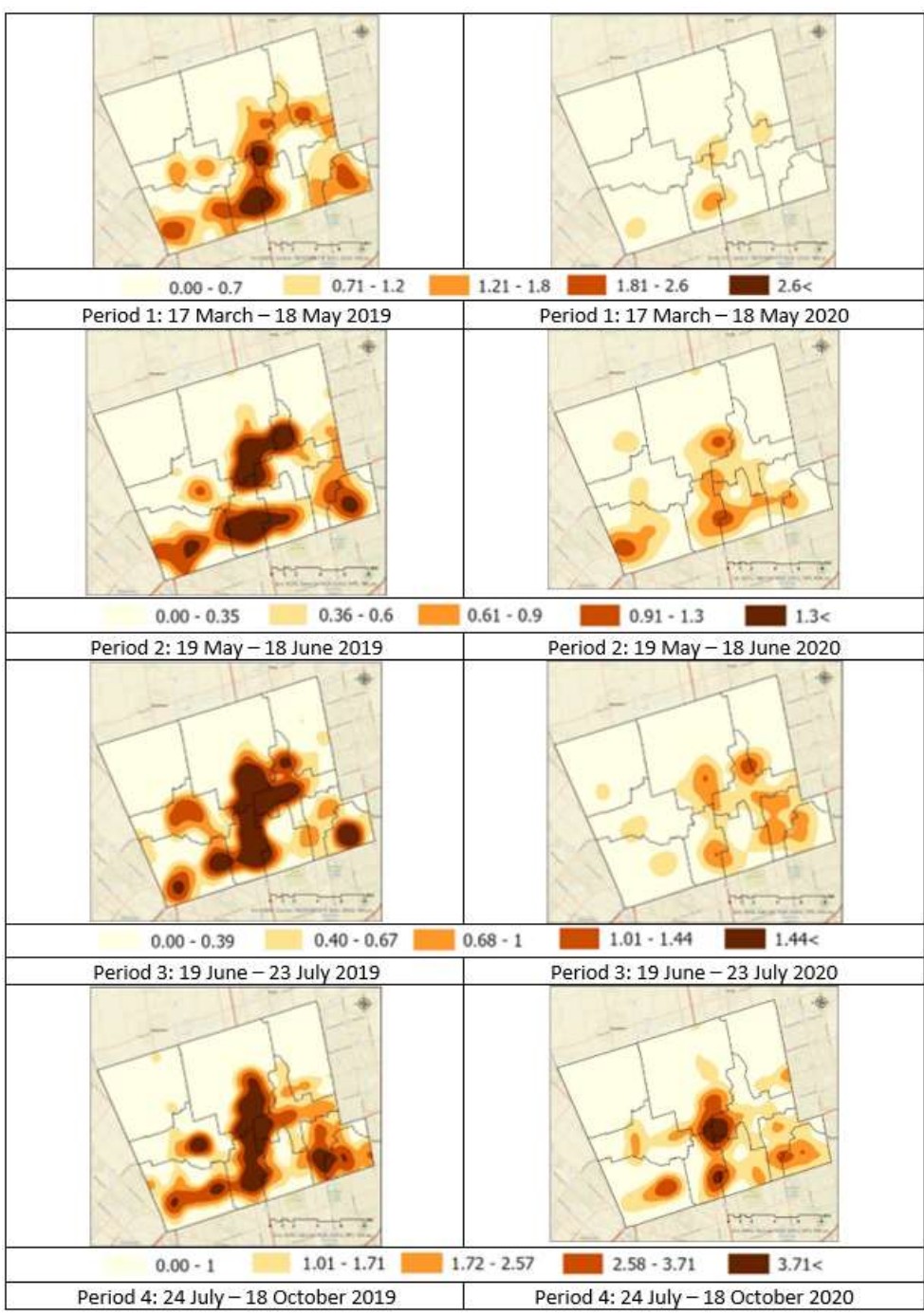

**Figure 8.** *Cont.*

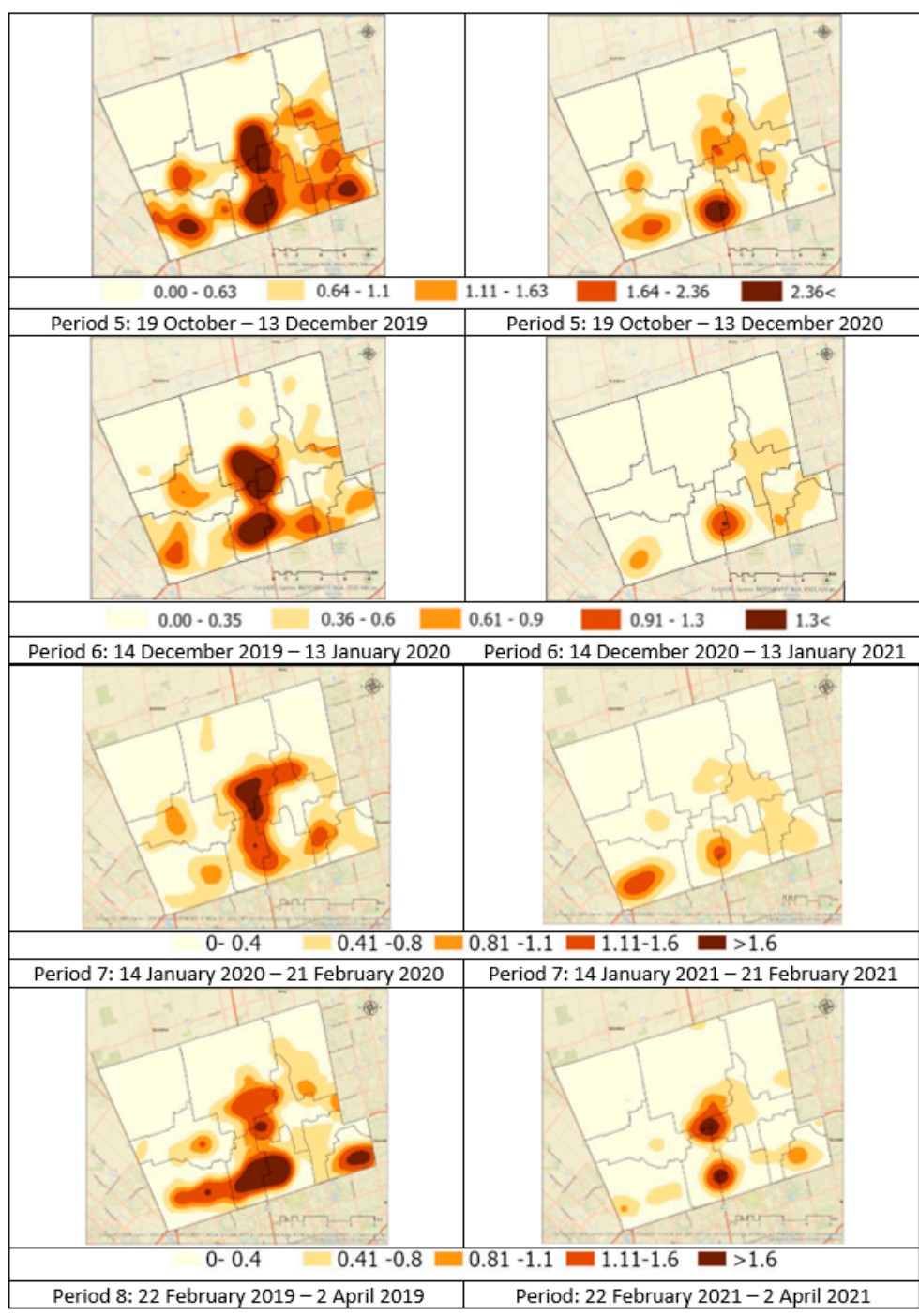

**Figure 8.** *Cont.*

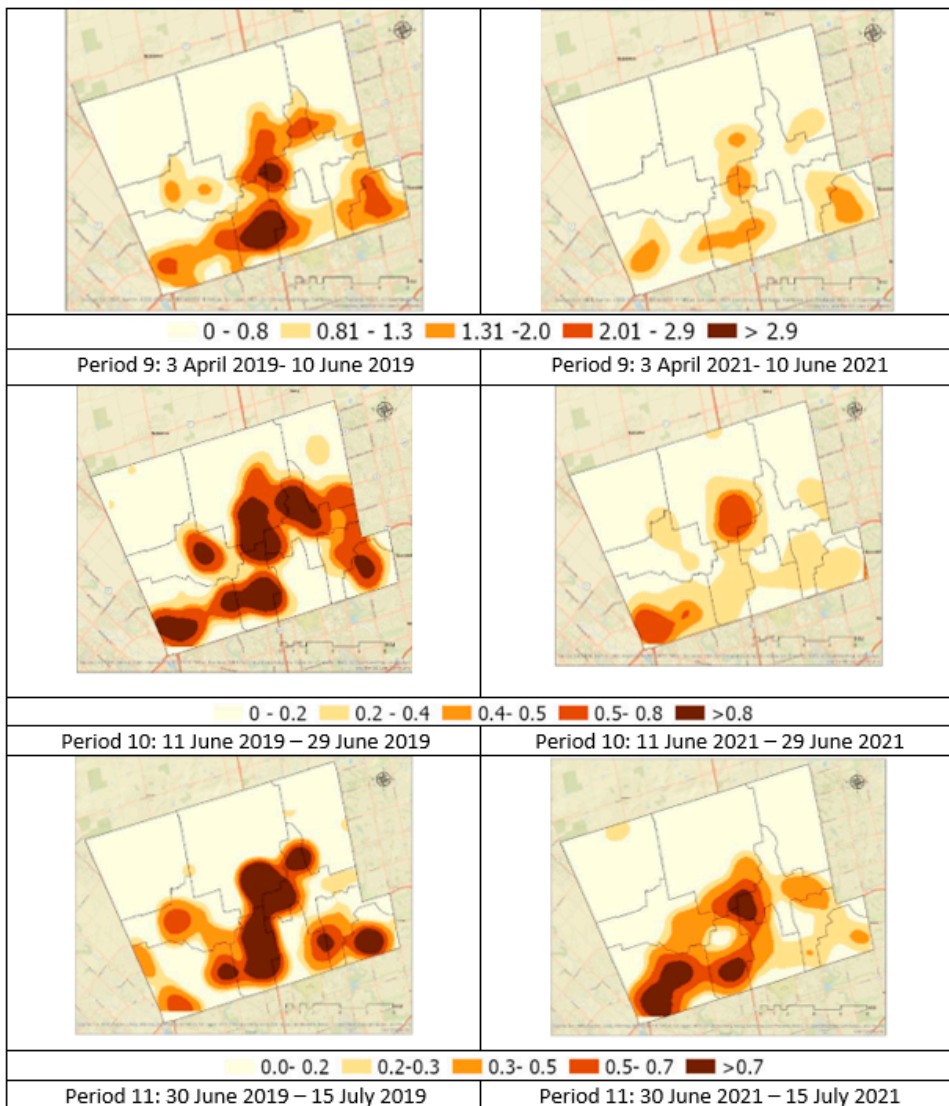

**Figure 8.** Kernel density maps for vehicle collisions/extrications for Periods 9–11.

### 3.1.4. False Fire Calls

False fire alarms constitute another major emergency call category. They may include alarm system equipment malfunctions or accidental or malicious fire alarm activations. Figure 9 illustrates the average number of false fire calls per day in each of Periods 1–11 in comparison with average daily numbers during the same period in the 3 years preceding the pandemic (2017–2019). Average numbers of false fire calls per day were lower during most COVID-19 periods compared to corresponding periods in 2017–2019, except apparently for Period 3 (Stage 2 of the reopening of the economy) and Period 11 (Step 2 of provincial reopening), when there were roughly comparable frequencies of false fire calls in relation to numbers during the corresponding pre-pandemic periods.

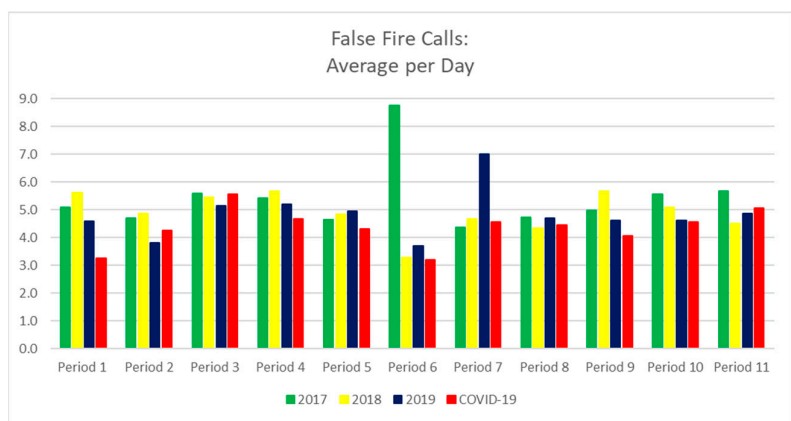

**Figure 9.** False fire calls: Average per day during each COVID-19 period versus corresponding periods in 2017, 2018, and 2019. Note: For Period 6, numbers reported for 2017, 2018, and 2019 are for 14 December 2017–13 January 2018, 14 December 2018–13 January 2019, and 14 December 2019–13 January 2020, respectively.

Kernel density maps for this type of calls are presented in Figure 10. These density maps show significant differences between the pandemic and non-pandemic periods and among different periods of the pandemic. For example, the density map shows an increase in density within Districts 77 and 73 compared to the previous year. As public health measures are relaxed and the economy reopens, the spatial pattern of false fire alarms becomes closer to the corresponding non-pandemic period.

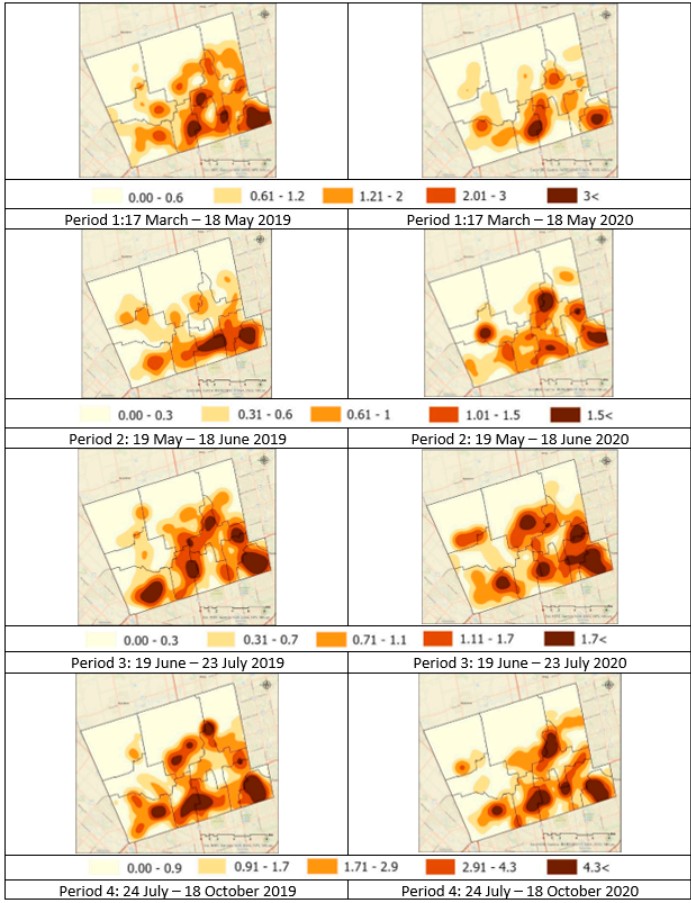

**Figure 10.** *Cont.*

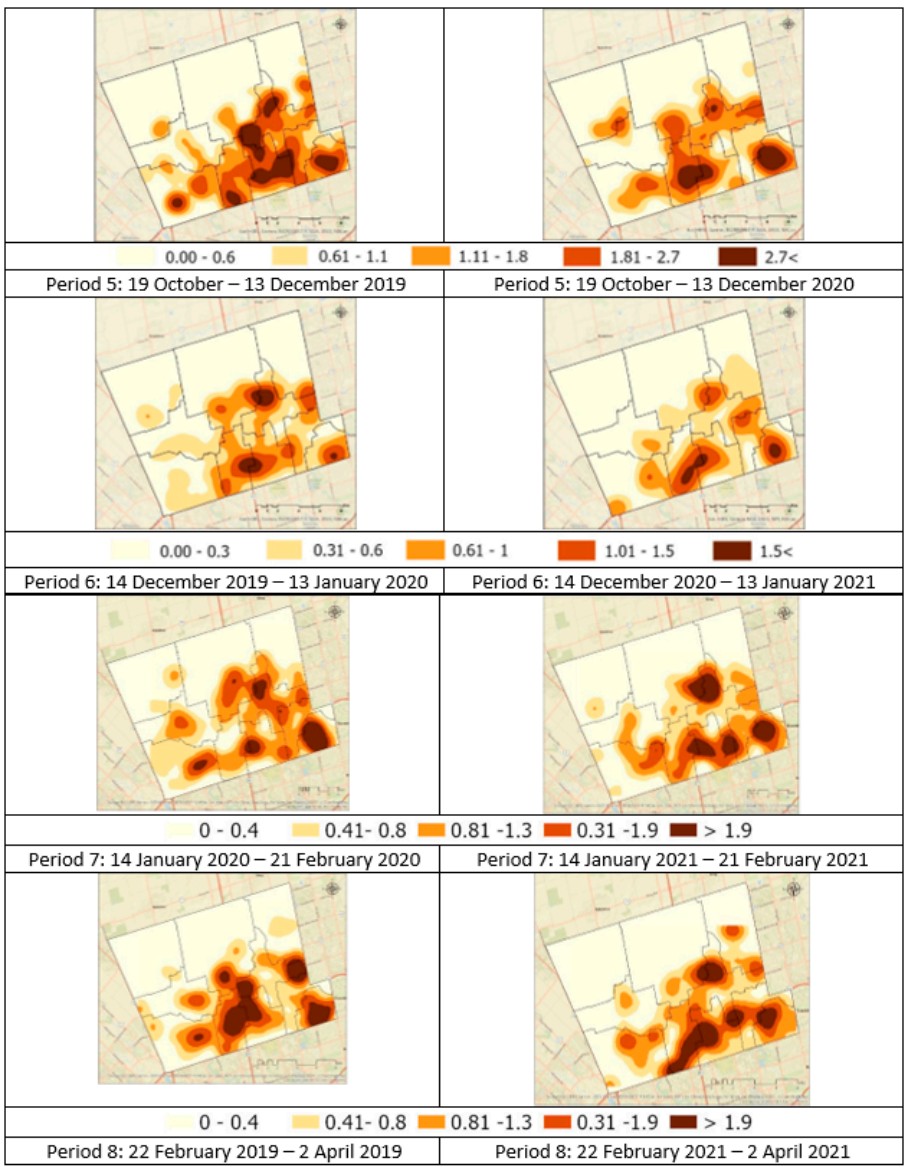

**Figure 10.** *Cont.*

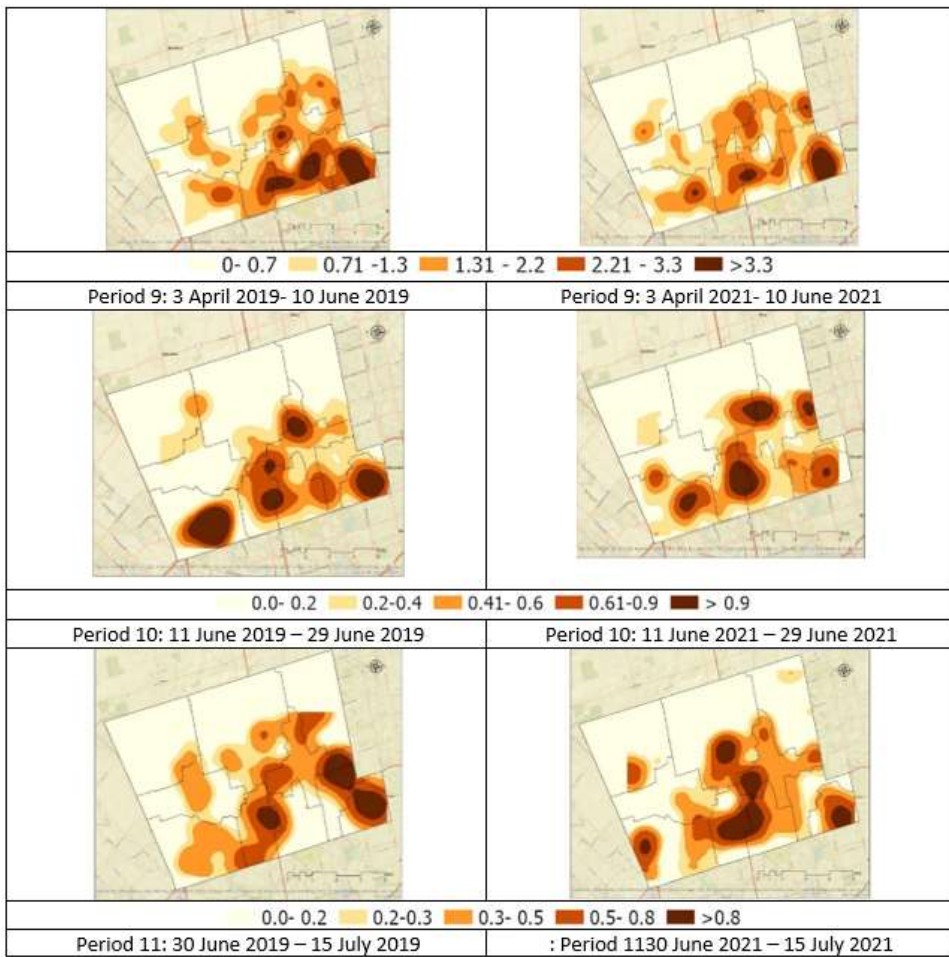

**Figure 10.** Kernel density maps for false fire calls for Periods 9–11.

*3.2. Emerging Hotspot Analysis*

The emerging hotspot analysis was carried out to identify patterns (as defined in Table 2) in emergency incidents that have occurred in the city of Vaughan, during the pandemic and pre-pandemic periods. Emergency incidents that occurred during 17 March 2019 to 16 March 2020 were considered as pre-pandemic, while incidents occurring from 17 March 2020 (the start of Period 1) onwards were considered as incidents during the pandemic period. Periods starting from 17 March 2019 and into the pandemic were also considered to capture the hotspots considering both pre-pandemic and pandemic periods.

The analysis was carried out using emergency incident data from 17 March 2019, which is 1 year prior to the start of Period 1 of the pandemic. The emergency incidents should be aggregated into defined locations and into defined time intervals to carry out the analysis. For this purpose, space–time cubes were created by aggregating points in ArcGIS Pro. The incidents were aggregated into a hexagonal grid with a distance interval of 1 km and a time interval of 2 weeks. The 'dispatch date' recorded for the incidents in the VFRS database was considered to aggregate data into time intervals. Created space–time cubes were then used to carry out the emerging hotspot analysis. The neighbourhood distance considered for the analysis was 2 km.

3.2.1. All Emergencies

The output maps from the emerging hotspot analysis provide a visual representation of changes in patterns of emergency calls in different parts of the City of Vaughan over space and time. The first map (first row, first column) in Figure 11 shows the hotspot patterns in the 1 year prior to the start of the pandemic (i.e., before a state of emergency

was declared and the very first lockdown began). During this time, we see that the persistent and oscillating hotspots are mainly located in Districts 71 and 75, and around the major highways of the city. These can be attributed to the presence of more densely populated residential neighbourhoods there compared to other areas, and more calls related to vehicular emergencies happening on the main highways. The different types of cold spots are prevalent in the northern and western parts of the City of Vaughan, which is primarily because those areas are far less populated and more rural, which decreases the chances of emergency calls. When this pattern is compared including Period 1, the time of the first lockdown imposed in the City of Vaughan, a marked decrease in the hotspots is seen (first row, second column of Figure 11). A lot of areas that previously used to be persistent hotspots changed to sporadic or diminishing hotspots, which is indicative of the fact that emergency calls during this period of the pandemic had decreased. The maps also show clear diminishing hotspots around the highways after restrictions were imposed, which is consistent with the fact that people were mostly staying home, thus causing a fall in vehicular emergency calls. Furthermore, we observe that as restrictions start to ease up (up to Period 4), more sporadic hotspots begin to appear, especially in Districts 73 and 75 (second row, first column of Figure 11). In comparison to this stage, in the map for the next few phases (i.e., until Period 11, when restrictions were slowly being lifted after months of fluctuating restrictions during the second and third waves of the pandemic), we see an increase in persistent hotspots in District 71, as well as some intensifying and sporadic hotspots in Districts 73 and 75 (second row, second column of Figure 11). It can be speculated that this increment was due to the higher frequencies of medical-related emergency calls, as COVID-19 case numbers were also significantly high during parts of this time. Overall, the pattern for cold spots remained fairly similar to that of pre-pandemic times.

### 3.2.2. Medical Emergencies

The results of EHA for medical emergency calls (Figure 12) are consistent with the other findings.

Persistent and oscillating hotspots are prevalent in the more densely populated parts of the city (particularly District 71). While these hotspots decreased during times when the numbers of COVID-19 cases were lower, more intensifying and new hotspots appeared after the case numbers started increasing again. It can be assumed that during that time, more people experiencing symptoms of the novel coronavirus or similar symptoms were likely to make medical-related emergency calls. In addition, an increase in the number of cold spots (mostly sporadic or persistent) for medical emergencies is observed during the pandemic in the non-residential/rural (e.g., District 74) or commercial/industrial (e.g., Districts 73 and 76) areas. This may be attributed to the fact that, with restrictions imposed and more people working from home, there were far fewer people than usual in those areas.

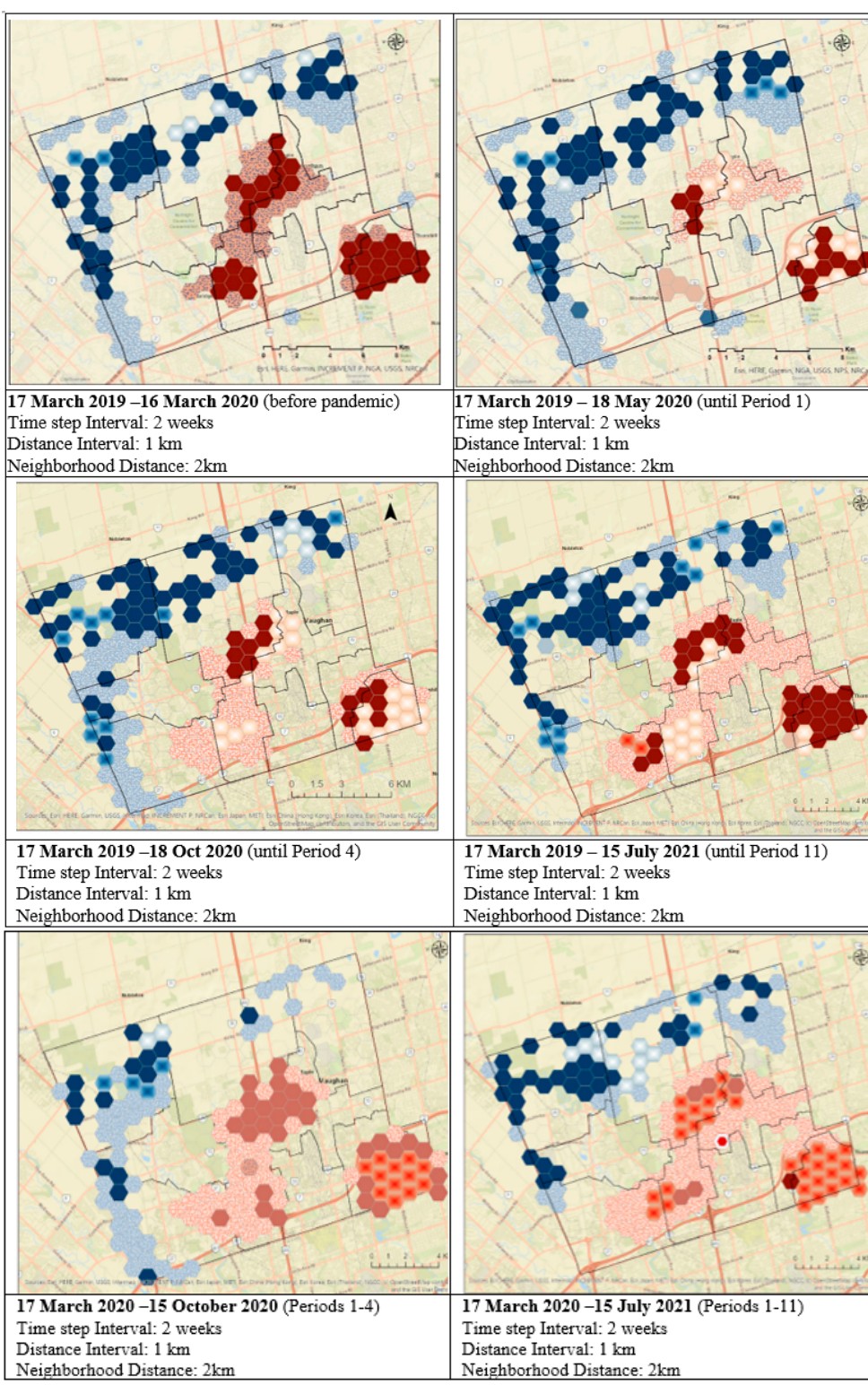

**Figure 11.** Emerging hotspot analysis before and during the pandemic for all emergency calls in the City of Vaughan.

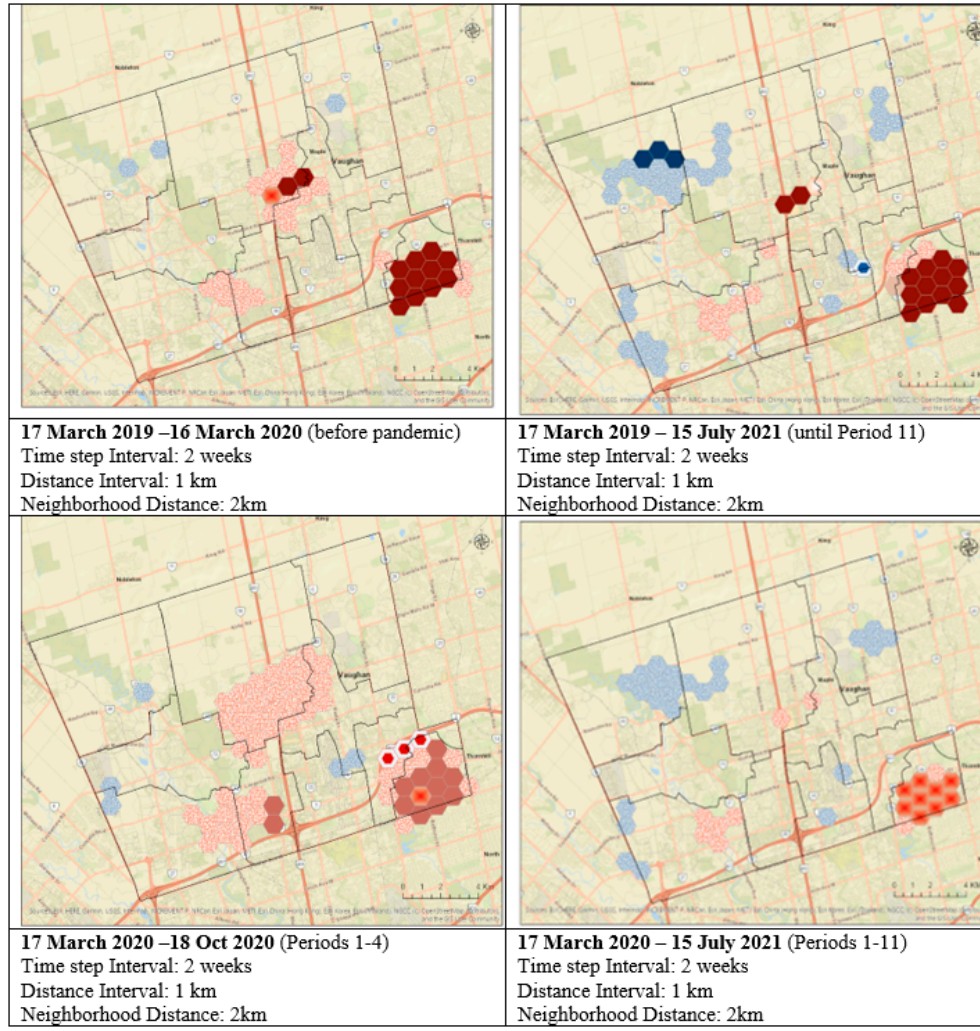

**Figure 12.** Emerging hotspot analysis before and during the pandemic for medical emergency calls in the City of Vaughan.

### 3.2.3. Vehicle Collisions/Extrications

Figure 13 presents the maps consisting of vehicular emergency calls. These findings show that the sporadic hotspots were concentrated mainly around the major highways (Ontario Highways 7, 400, and 407), which cut through the City of Vaughan. It can be inferred that, during periods of lockdowns and restrictions, there were far fewer vehicles on the roads compared to pre-pandemic times, resulting in fewer hotspots. On the contrary, when restrictions were being lifted and there was more movement of people on the roads, hotspots begin to develop again. Thus, the fluctuating phases of restrictions leave us with sporadic hotspots for vehicular emergency calls. An interesting observation is increased sporadic, consecutive, and new hotspots for vehicle emergency calls in District 73, near Highways 7 and 407. While before the pandemic there seems to be no pattern of hot or cold spots in this area, an increased appearance of hotspots was seen in the emerging hotspot analysis of the time during the pandemic (17 March 2020–15 July 2021). This may, however, be associated with the relatively darker kernel density map for Period 11 in Figure 11.

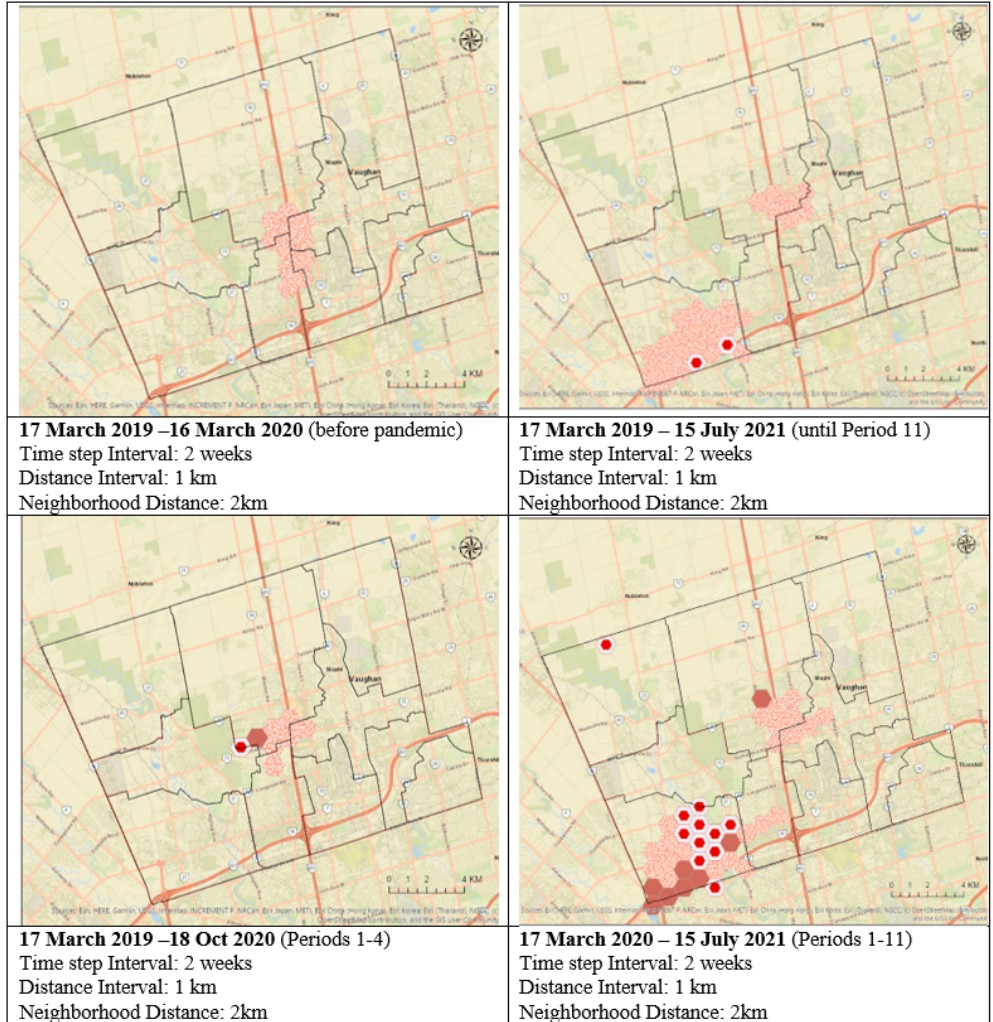

**Figure 13.** Emerging hotspot analysis before and during the pandemic for vehicle collisions/extrications in the City of Vaughan.

## 4. Discussion

The COVID-19 pandemic interrupted societal functions and activities and demand for municipal services were impacted as a result of that as well. Emergency calls that are one of the key services that cities provide were particularly impacted in many different ways. In this study, we applied spatial and spatiotemporal analyses to demonstrate the usefulness of these tools in showing the changing patterns of emergency calls during the COVID-19 pandemic. Once the data had location and geographic coordinates, these methods could be quickly and continuously applied to derive more information about the ongoing impacts of the pandemic on emergency calls or other similar data.

While spatiotemporal changes in the distribution of emergency calls during the pandemic were expected, this study provides some insights on how these patterns are reflected in data. Our analysis, covering a significant part of the COVID-19 pandemic with different waves and phases of public health measures, is able to detect the spatiotemporal changes. The findings can be used to further study potential factors that can explain the changes as well as to make the necessary planning for emergency services for the current and future pandemics.

Another important aspect of these trends is that a reduction in emergency calls caused by daily emergencies can reduce pressure on health care systems during the pandemic, when hospitals and health care workers are busy dealing with the infected individuals who are hospitalized. Knowledge of these patterns and their sensitivity to implemented public

health measures may be used to guide resource allocation and be considered during the pandemic response planning.

The spatiotemporal analysis clearly showed changes in the distribution of emergency calls in time and space. Of course, the change in patterns reflects both changes in the total number of emergency calls on one hand and shifting of the numbers between different spaces on the other hand. Therefore, districts with less diverse land uses experience larger changes compared to districts with mixed land uses. This implies that fire stations serving such districts will be impacted differently.

While the long-term impacts of the pandemic are yet to be known, it can be argued that if a significant share of workers continues to work from home, this could also impact the emergency calls in the long term.

Overall, exploring and understanding these changes in patterns of hotspots and cold spots can aid in better planning and allocating of resources for emergency calls in the City of Vaughan for similar scenarios. For instance, if newer hotspots or intensifying hotspots are appearing, the authorities can decide to concentrate more resources in those areas. Similarly, diminishing hotspots or cold spots can inform the authorities to redirect or shift resources from those areas into other areas of pressing concern.

The emergency call hotspots and cold spots, respectively, show areas with high and low emergency service demand and their changes over time. It is important to note here that understanding both cold and hotspots is important, as they each have different implications in terms of use and need for resources. Considering the needs and situations, resource management can be adjusted based on the spatiotemporal trends.

Moreover, since the pandemic has several waves, this information can shed light into what will be expected to happen should public health agencies and federal/provincial governments need to increase or decrease public health measures.

It is important to note that this study is limited in a number of ways: (1) this study only covered the first 16 months of the pandemic in the City of Vaughan (17 March 2020–15 July 2021). Although this period was the most difficult part of the pandemic and before the mass vaccination, patterns may have changed significantly after the relaxation of the public health measures during the following phases; and (2) this study did not examine the contributing factors into the observed and extracted trends. Future studies can examine in more detail the relationship between various local factors that can explain these patterns.

## 5. Conclusions

This study has examined the geographic distributions and spatiotemporal patterns of emergency calls in the City of Vaughan during the first 11 periods of the COVID-19 pandemic and compared them with corresponding pre-pandemic periods in 2017–2019. We applied the kernel density analysis and emerging hot and cold spot analyses over the 11 periods of the pandemic, with each period related to specific levels of public health measures/restrictions.

The results suggest that the COVID-19 pandemic and public health measures introduced to respond to it during different periods had significant impacts on the spatiotemporal distribution of emergency calls in the city. These may have potential implications for resource planning and allocation across the city's fire districts/stations. They could also provide insights on how to manage fire and rescue service operations as further stages of the pandemic unfold.

While conventional methods and analyses can show such changes to some extent, spatiotemporal analyses enable relating these changes in space over time to further examine locational attributes that determine changing patterns in occurrences of emergency incidents.

Emergency service decision makers can apply insights gained from the analyses in the planning and management of limited resources. Closure of schools, restaurants, and non-essential businesses and stay-at-home orders shift activities from buildings and public spaces to homes and, depending on land use patterns and distribution of service

centres, this can impact how emergency services are provided, particularly with respect to reallocating the firefighting apparatus and crews to the various fire districts/stations according to the emerging situations.

**Author Contributions:** Conceptualization, A.A. and A.O.S. and M.S.S.; methodology, A.A., A.O.S., J.W. and N.K., temporal and spatial data preparation, A.A., A.O.S., J.W. and N.K.; formal analysis, A.A., J.W., N.K. and A.O.S., writing—original draft preparation, J.W., N.K. and A.O.S.; writing—review and editing, A.A., A.O.S. and M.S.S.; funding acquisition, A.O.S. and A.A. All authors have read and agreed to the published version of the manuscript.

**Funding:** This research has been conducted with financial support from the Social Sciences and Humanities Research Council of Canada (SSHRC) as part of its Partnership Engage Grants (PEG) COVID-19 Special Initiative, under grant number 1008-2020-0217. The Vaughan Fire and Rescue Service (VFRS) is the partner organization of the York University research team in this effort. The funders had no role in study design, data collection and analysis, decision to publish, or preparation of the manuscript.

**Data Availability Statement:** Original data for this research was obtained from VFRS through an agreement with York University and cannot be shared.

**Conflicts of Interest:** The authors declare no conflict of interest.

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
