# Peer review of "Spatiotemporal Analysis of Emergency Calls during the COVID-19 Pandemic: Case of the City of Vaughan"

_urbansci, doi:10.3390/urbansci7020062_

Round 1

Reviewer 1 Report

1.     What are the contributions of this study? The authors need to clarify the contributions in the Introduction Section.

2.     The manuscript lacks reviewing the existing studies and methods. As such, the description related work is needed to be added.

3.     Line 92-93, “These 11 periods are based upon public health measures introduced in the province of Ontario and in York Region.” Please add a reference here.

4.     In Section 2.3, an overall workflow is needed to be provided to make the methodology and overall process clearer.

5.     In lines 79-80, “VFRS provided the dataset of all occurrences of incidents within the City of Vaughan for the period of 01 January 2017 to 15 July 2021.” Please add a reference here to indicate where the data can be downloaded.

It is good. 

Author Response

AUTHORS’ RESPONSES TO REVIEWER 1’S COMMENTS

Quality of English Language

 (x) Minor editing of English language required

Yes

Can be improved

Must be improved

Not applicable

Is the content succinctly described and contextualized with respect to previous and present theoretical background and empirical research (if applicable) on the topic?

( )

(x)

( )

( )

Are all the cited references relevant to the research?

( )

(x)

( )

( )

Are the research design, questions, hypotheses and methods clearly stated?

( )

(x)

( )

( )

Are the arguments and discussion of findings coherent, balanced and compelling?

( )

(x)

( )

( )

For empirical research, are the results clearly presented?

( )

(x)

( )

( )

Is the article adequately referenced?

( )

( )

(x)

( )

Are the conclusions thoroughly supported by the results presented in the article or referenced in secondary literature?

( )

(x)

( )

( )

Comments and Suggestions for Authors

Reviewer’s Comment

  1. What are the contributions of this study? The authors need to clarify the contributions in the Introduction Section.

Our Response

Agree with the reviewer’s comment. We have further explained the contribution of the paper in the introduction.

Reviewer’s Comment

  1. The manuscript lacks reviewing the existing studies and methods. As such, the description related work is needed to be added.

Our Response

We have added brief descriptions of related work as suggested.

Spatiotemporal analysis can play a vital role in understanding and analysing data with spatial attributes. Spatiotemporal modelling methodologies are rapidly developing and evolving. Emerging Hotspot Analysis (EHA) is among the new methods added to the GIS based analyses and its usage in spatiotemporal analysis is growing [1]–[7]. Gudes et al. [1] used spatial modelling and spatiotemporal methods to identify emerging hotspots of heavy-vehicle crashes on specific roads in Western Australia. Rabiei-Dastjerdi and McArdle [2] investigated patterns of neighbourhood change by using EHA of Airbnb data in the City of Dublin, Ireland. Using EHA, Reddy et al. [3] found the dominance of sporadic hotspots and persistent hotspots in vegetation fire occurrences in Myanmar and South Asian countries. EHA appears to have been used prevalently in forecasting crimes – e.g., by Hart [4] to forecast crime hotspots for three types of crime handled by six U.S. law enforcement agencies, by Adepeju et al. [5] for three crime types in the borough of Camden (London, UK) and South Chicago (U.S.), by Mohler [6] for homicide and gun crimes in Chicago (U.S.), and by Chainey et al. [7] for four crime types.

Spatial and spatiotemporal analysis tools have become indispensable in studying public health concerns during the COVID-19 pandemic. A number of studies have attempted to explore the spatial and spatiotemporal patterns, including EHA, of COVID-19 cases and underlying risk factors of COVID-19 in different contexts [8]–[11]. Mollalo et al. [8] applied spatial modeling tools, including multiscale geographically weighted regression (MGWR) analysis, to the county-level counts of COVID-19 cases from January 22 – April 9, 2020 across the continental U.S. They found that MGWR could explain 68% of the total variations of COVID-19 incidence. Mylona et al. [9] extracted and performed hotspot analysis of influenza cases (2016-2019) as well as COVID-19 cases (March-April 2020) from a Rhode Island (U.S.) hospital network to simulate a real-time surveillance scenario. Andersen et al. [10] analyzed spatial determinants of local Covid-19 transmission in the U.S. and found COVID-19 hotspots predominantly in New England, Southeast, and Southwest states. Purwanto et al. [11] conducted spatiotemporal analysis of COVID-19 spread in East Java, Indonesia, using EHA and space-time cube models. Results showed that the spread of COVID-19 in East Java was centered in Surabaya, then spread to suburban areas and other cities.

Reviewer’s Comment

  1. Line 92-93, “These 11 periods are based upon public health measures introduced in the province of Ontario and in York Region.” Please add a reference here.

Our Response

It is not possible to provide a single reference for the specification of all of these 11 periods. References were cited in the two paragraphs preceding Table 1. To summarize, however, we have modified Table 1 to show Reference(s) for each of Periods 1-11, as follows:

Reviewer’s Comment

  1. In Section 2.3, an overall workflow is needed to be provided to make the methodology and overall process clearer.

Our Response

Thank you very much for the suggestion. A workflow chart was added.

Reviewer’s Comment

  1. In lines 79-80, “VFRS provided the dataset of all occurrences of incidents within the City of Vaughan for the period of 01 January 2017 to 15 July 2021.” Please add a reference here to indicate where the data can be downloaded.

We have obtained the data through a Non-Disclosure Agreement and are not permitted to publish or provide access to the raw data. Those interested in getting the data can contact the City of Vaughan Fire and Rescue Service. To make this clear we added a sentence in the data section.

Reviewer’s Comment on the Quality of English Language

It is good. 

Our Response

We sincerely appreciate this reviewer’s assessment of the quality of English language. We have, nonetheless, sought to introduce edits/improvements in this paper revision, where appropriate.

Reviewer 2 Report

This paper  deals with an important topic for the reality of contemporary cities: emergency care during the Covid 19 pandemic and its impacts in the city. Using the city of Vaughan, Canada, as a case study, the paper presents a research that examined the geographic distributions and spatiotemporal patterns of emergency calls obtained from the Vaughan Fire and Rescue Service (VFRS).   

The authors considered data from VFRS during the first 11 periods of the  COVID-19 pandemic (March 2020 to July 2021), subdivided according to restriction levels in the city and they  compared with the corresponding data from pre-pandemic periods (2017-2019). 

The study presents an innovative methodology: analysis of kernel density (space-time) and emerging hotspot, that is, analysis of emerging hot and cold spots in different periods of the pandemic, related to specific levels of public health measures/restrictions. 

The research demonstrates the differentiated behavior of public health measures that are stricter or not and of densities and housing services in certain central neighborhoods to the detriment of other more peripheral ones, such as lockdowns and closure of non-essential businesses. According to the authors, this is a first-of-its-kind study that applies spatiotemporal methods to assess changes in the frequency and mix of emergency incidents that were attended to by a municipal fire and rescue service in different periods of the pandemic. 

The results suggest that the COVID-19 pandemic and the public health measures instituted to respond to it in different periods had significant impacts on the distribution of emergency calls in the city space.  

The importance of the study indicates that the results can provide useful information both for resource management in emergency services and for understanding the underlying causes of such patterns. 

Despite the importance of the study and the method, the article does not make clear the main questions and research hypothesis.  

This type of study can be considered a decision-making tool to plan and allocate resources in emergency care sectors, as the authors point out. However, it would be important to highlight the hypothesis of the study at the beginning of this paper and reinforce the relationship between the emergency service during the different phases of the Pandemic as well as the patterns of density of urban space. 

 I recommend its publication, as long as the hypothesis that guides the research is explained.

Author Response

AUTHORS’ RESPONSES TO REVIEWER 2’S COMMENTS

Quality of English Language

(x) I am not qualified to assess the quality of English in this paper

Yes

Can be improved

Must be improved

Not applicable

Is the content succinctly described and contextualized with respect to previous and present theoretical background and empirical research (if applicable) on the topic?

(x)

( )

( )

( )

Are all the cited references relevant to the research?

(x)

( )

( )

( )

Are the research design, questions, hypotheses and methods clearly stated?

( )

(x)

( )

( )

Are the arguments and discussion of findings coherent, balanced and compelling?

(x)

( )

( )

( )

For empirical research, are the results clearly presented?

(x)

( )

( )

( )

Is the article adequately referenced?

(x)

( )

( )

( )

Are the conclusions thoroughly supported by the results presented in the article or referenced in secondary literature?

(x)

( )

( )

( )

Comments and Suggestions for Authors

Reviewer’s Comment

This paper  deals with an important topic for the reality of contemporary cities: emergency care during the Covid 19 pandemic and its impacts in the city. Using the city of Vaughan, Canada, as a case study, the paper presents a research that examined the geographic distributions and spatiotemporal patterns of emergency calls obtained from the Vaughan Fire and Rescue Service (VFRS).   

The authors considered data from VFRS during the first 11 periods of the  COVID-19 pandemic (March 2020 to July 2021), subdivided according to restriction levels in the city and they  compared with the corresponding data from pre-pandemic periods (2017-2019). 

The study presents an innovative methodology: analysis of kernel density (space-time) and emerging hotspot, that is, analysis of emerging hot and cold spots in different periods of the pandemic, related to specific levels of public health measures/restrictions. 

The research demonstrates the differentiated behavior of public health measures that are stricter or not and of densities and housing services in certain central neighborhoods to the detriment of other more peripheral ones, such as lockdowns and closure of non-essential businesses. According to the authors, this is a first-of-its-kind study that applies spatiotemporal methods to assess changes in the frequency and mix of emergency incidents that were attended to by a municipal fire and rescue service in different periods of the pandemic. 

The results suggest that the COVID-19 pandemic and the public health measures instituted to respond to it in different periods had significant impacts on the distribution of emergency calls in the city space.

The importance of the study indicates that the results can provide useful information both for resource management in emergency services and for understanding the underlying causes of such patterns. 

Our Response

Thank you very much for the feedback.

Reviewer’s Comment

Despite the importance of the study and the method, the article does not make clear the main questions and research hypothesis.  

This type of study can be considered a decision-making tool to plan and allocate resources in emergency care sectors, as the authors point out. However, it would be important to highlight the hypothesis of the study at the beginning of this paper and reinforce the relationship between the emergency service during the different phases of the Pandemic as well as the patterns of density of urban space. 

 I recommend its publication, as long as the hypothesis that guides the research is explained.

Our Response

Thank you very much for the comment. As an exploratory/descriptive and applied study this work was not primarily a hypothesis driven research. We did not aim to explicitly test any hypothesis. We were only trying to understand the patterns. It is our next research to examine in detail the relationship between the observed patterns and some of the key local factors. However, to address reviewers’ request we added some key questions in the last paragraph of the introduction to better guide the intention of our study (lines 91-102.)

“In particular, we were interested in finding answers to the following questions: how has pandemic changed the emergency calls in the city? Has this pattern changed during different phases of the pandemic under different public health measures? Which emergency calls had the highest and lowers spatial-temporal changes?”

Reviewer 3 Report

Thank you for inviting me to review this article.

The topic of the paper is of great interest considering the impact that COvid had on society as a whole. The results obtained contribute to generating knowledge about emergency call patterns in the city of Vaughan, following the application of different spatial techniques. Furthermore, the authors point out that these results can help decision-makers to better manage resources. In this sense, the authors should rethink the objective, including the applied aspects of the work that should be addressed in the discussion. This would prevent the article from being so descriptive.

Although the article notes that this is the first study to examine the spatio-temporal patterns of emergency calls, it should better contextualise the phenomenon studied in the introductory section: the organisation of emergency services in cities.

In addition, the authors should justify why they have chosen the city of Vaughan as an example.

The analysis techniques used and the cartographic representation visually express a reality that needs interpretative comment.

Finally, the conclusions should reflect the possible limitations of the work and be more critical.

Author Response

AUTHORS’ RESPONSES TO REVIEWER 3’S COMMENTS

Quality of English Language

(x) I am not qualified to assess the quality of English in this paper

Yes

Can be improved

Must be improved

Not applicable

Is the content succinctly described and contextualized with respect to previous and present theoretical background and empirical research (if applicable) on the topic?

( )

( )

(x)

( )

Are all the cited references relevant to the research?

( )

(x)

( )

( )

Are the research design, questions, hypotheses and methods clearly stated?

( )

( )

(x)

( )

Are the arguments and discussion of findings coherent, balanced and compelling?

( )

(x)

( )

( )

For empirical research, are the results clearly presented?

(x)

( )

( )

( )

Is the article adequately referenced?

( )

(x)

( )

( )

Are the conclusions thoroughly supported by the results presented in the article or referenced in secondary literature?

( )

( )

(x)

( )

Comments and Suggestions for Authors

Thank you for inviting me to review this article.

Reviewer’s Comment

The topic of the paper is of great interest considering the impact that COvid had on society as a whole. The results obtained contribute to generating knowledge about emergency call patterns in the city of Vaughan, following the application of different spatial techniques. Furthermore, the authors point out that these results can help decision-makers to better manage resources. In this sense, the authors should rethink the objective, including the applied aspects of the work that should be addressed in the discussion. This would prevent the article from being so descriptive.

Our Response

We agree with the reviewer. We have added some sentences to the introduction and have expanded our discussion to address the reviewer’s comment. All the changes in these sections have been highlighted by different color and tracked.

Reviewer’s Comment

Although the article notes that this is the first study to examine the spatio-temporal patterns of emergency calls, it should better contextualise the phenomenon studied in the introductory section: the organisation of emergency services in cities.

Our Response

We thank the reviewer for this useful and insightful comment/suggestion. We have added the following paragraph in section 1 (Introduction): 

In cities and municipalities in the province of Ontario, Canada, emergency calls that are responded to by fire departments, referred to in certain cases as fire and rescue services, are not restricted to fires or fire-related emergencies. Emergency incidents fall under a number of major categories other than property fires/explosions, including false fire calls, medical emergencies, vehicle collisions/extrications, public hazards (including carbon monoxide), and others. Emergency incidents falling under various categories will call for different resource requirements – e.g., number and types of responding vehicles, number and training/experience of crew members. The geographical expanse of a city/municipality and the distribution of properties across property type (e.g., residential, business and personal services, industrial, mercantile, assembly, care and detention, vehicles, etc.) would usually call for the subdivision of the city/municipality into fire districts and the distribution or allocation of firefighting and rescue vehicles/apparatus across different fire stations.

We have further added the following two paragraphs in subsection 2.1 (Study Area), accompanied by a new Figure 2a:

As of January 2020, two responding units – where the term ‘responding unit’ refers to firefighting apparatus manned by a crew of four firefighters – were stationed at each of VFRS’ Stations 7-1, 7-2, and 7-3. The seven other fire stations each have only one responding unit. However, in addition to a full responding unit, Station 7-5 also has a first response unit, which has a crew of at least two firefighters using a pick-up truck, mostly responding to medical emergency calls. A limited number of specialized firefighting apparatus are based at designated fire stations. For example, engines having 100 ft. aerial equipment are based at Stations 7-1 and 7-3, with more high-rise buildings located in Districts 71 and 73 than in other districts.

In 2019, the calendar year immediately preceding the declaration of the COVID-19 pandemic, the VFRS responded to a total of 11,313 emergency incidents of various types (see Figure 2a). Medical emergencies accounted for 47% of all emergency incidents, followed by false fire calls (16.1%) and vehicle collisions/extrications (14.1%), There were only 263 property fires/explosions, accounting for only 2.3% of all emergency incidents in 2019.

We trust that the above additions would serve to further contextualize the need to examine spatiotemporal patterns of emergency calls.

Reviewer’s Comment

In addition, the authors should justify why they have chosen the city of Vaughan as an example.

Our Response

With respect to this very relevant comment, we will include the following information in the statement on Funding in the final manuscript:

Funding: This research has been conducted with financial support from the Social Sciences and Humanities Research Council of Canada (SSHRC) as part of its Partnership Engage Grants (PEG) COVID-19 Special Initiative. The Vaughan Fire and Rescue Service (VFRS) is the partner organization of the York University research team in this effort.

The two lead authors/project investigators (Asgary and Solis) had undertaken earlier research on VFRS pre-pandemic operations. In the earlier work, simulation studies have provided insights into how to allocate VFRS’ resources (firefighting and rescue apparatus, response vehicles, and crews) across fire districts and stations. This work was incorporated into a collaborative project with VFRS entitled ‘Igniting Insight: Using GIS and Analytics in the Fire Service’. In recognition of this collaborative project, VFRS was given the 2019 Innovative Management Bronze Award by the Institute of Public Administration of Canada. The successful prior collaboration with VFRS by the lead authors, therefore, have rendered more readily attainable the application of various data analytics methods to VFRS incident and response data prior to and during the COVID-19 pandemic.

Reviewer’s Comment

The analysis techniques used, and the cartographic representation visually express a reality that needs interpretative comment.

Our Response

We were not very sure about the comment. We have not interpreted the visuals on their caption as we have explained or interpreted them in the relevant text section.

Reviewer’s Comment

Finally, the conclusions should reflect the possible limitations of the work and be more critical.

Our Response

Agreed. We added the studies limitations at the end of discussion section.

Reviewer 4 Report

Although this is a very interesting paper, there are still many areas that can be improved. Here are my comments:

1. The introduction part lacks a significant literature review, making it impossible to support the authors' study questions and the content of the study conducted.

2. All the figures in the study are strange, more like screenshots than software drawings.

3. The methods involved should be clearly outlined, including formulas and other technical details. So it is suggested that the author draw a technical diagram of the study.

4. The result part needs to be strengthened.

5. There are too few references, so it is recommended to supplement them.

Minor editing of English language required

Author Response

AUTHORS’ RESPONSES TO REVIEWER 4’S COMMENTS

Quality of English Language

 (x) Minor editing of English language required

Yes

Can be improved

Must be improved

Not applicable

Is the content succinctly described and contextualized with respect to previous and present theoretical background and empirical research (if applicable) on the topic?

( )

(x)

( )

( )

Are all the cited references relevant to the research?

( )

( )

(x)

( )

Are the research design, questions, hypotheses and methods clearly stated?

( )

( )

(x)

( )

Are the arguments and discussion of findings coherent, balanced and compelling?

( )

( )

(x)

( )

For empirical research, are the results clearly presented?

( )

(x)

( )

( )

Is the article adequately referenced?

( )

(x)

( )

( )

Are the conclusions thoroughly supported by the results presented in the article or referenced in secondary literature?

( )

( )

(x)

( )

Comments and Suggestions for Authors

Reviewer’s Comment

Although this is a very interesting paper, there are still many areas that can be improved. Here are my comments: 

Our Response

Thank you very much for the general feedback. We have tried to improve the paper based on the great comments provided by you and other reviewers.

Reviewer’s Comment

  1. The introduction part lacks a significant literature review, making it impossible to support the authors' study questions and the content of the study conducted.

Our Response

Thanks for the comment. Agreed. We have made some significant changes to address the reviewer’s comment.

Reviewer’s Comment

  1. All the figures in the study are strange, more like screenshots than software drawings.

Our Response

The figures are screenshots of the software drawing. However, because we have so many figures for different phases and it is important to show them side by side they had to be reduced to smaller sizes.

Reviewer’s Comment

  1. The methods involved should be clearly outlined, including formulas and other technical details. So it is suggested that the author draw a technical diagram of the study.

Our Response

Agreed. We added a paragraph and figures at the beginning of section 2 to address this comment. We did not include the formulas in the paper because we though they are well-known and providing a reference to the relevant ArcGIS Pro guide would be sufficient. Thank you.

Reviewer’s Comment

  1. The result part needs to be strengthened.

Our Response

Not too sure which section is meant by the result section as we do not have a specific part called Results. We assumed that the reviewer is referring to the conclusion section. In that case we agree and expanded the conclusion section.

Reviewer’s Comment

  1. There are too few references, so it is recommended to supplement them.

Our Response

Agreed. We added some new references as we expanded our literature review. Thank you very much for the suggestion.

Reviewer’s Comments on the Quality of English Language

Minor editing of English language required.

Our Response

We acknowledge the reviewer’s call for “[minor editing of English language” and have sought to improve the English grammar and composition in this paper revision.

Round 2

Reviewer 3 Report

The authors have done an adequate review of the article. Thanks

Reviewer 4 Report

Accept in present form

Accept in present form